# The role of ice-sheet topography in the Alpine hydro-climate at glacial times

Patricio Velasquez[1,2,3], Martina Messmer[1,2], and Christoph C. Raible[1,2]

[1]Climate and Environmental Physics, Physics Institute, University of Bern, Bern, Switzerland
[2]Oeschger Center for Climate Change Research, University of Bern, Bern, Switzerland
[3]Institute for Atmospheric and Climate Science, ETH Zurich, Zurich, Switzerland

**Correspondence:** Patricio Velasquez (patricio.velasquez@env.ethz.ch)

**Abstract.** In this study, we investigate the sensitivity of the glacial Alpine hydro-climate to northern hemispheric and local ice-sheet changes. Bridging the scale gap by using a chain of global and regional climate models, we perform sensitivity simulations of up to 2 km horizontal resolution over the Alps for the Last Glacial Maximum (LGM) and the Marine Isotope Stage 4 (MIS4). In winter, we find wetter conditions in the southern part of the Alps during LGM compared to present day, to which dynamical processes, i.e. changes in the wind speed and direction, substantially contribute. During summer, we find the expected drier conditions in most of the Alpine region during LGM, as thermodynamics suggests drier conditions under lower temperatures. The MIS4 climate shows enhanced winter precipitation compared to the LGM, which is explained by its warmer climate compared to the LGM — thus, again explained by thermodynamics. The sensitivity simulations of the northern hemispheric ice-sheet changes show that an increase of the ice-sheet thickness leads to a significant intensification of glacial Alpine hydro-climate conditions, which is mainly explained by dynamical processes. Changing only the Fennoscandian ice sheet is less influential on the Alpine precipitation, whereas modifications in the local Alpine ice-sheet topography significantly alter the Alpine precipitation, in particular we find a reduction of summer precipitation at the southern face of the Alps when lowering the Alpine ice sheet. The findings demonstrate that the northern hemispheric and local ice-sheet topography play an important role in regulating the Alpine hydro-climate and thus permits a better understanding of the precipitation patterns in the complex Alpine terrain at glacial times.

## 1 Introduction

Glacial times are characterised by very different boundary conditions than today, leading to cold conditions, substantial sea-level drops and a strong increase in land ice sheets (Mix et al., 2001). This different climate behaviour of glacial times have attracted the scientific community, since they are an ideal test bed to challenge state-of-the-art climate models in their ability to simulate changes in climate (e.g. Kageyama et al., 2021). Further, glacial times are also suitable to identify relevant mechanisms such as feedback processes in the climate system (e.g. Stocker and Johnsen, 2003) and to investigate response behaviour to external forcing (e.g. Ganopolski and Calov, 2011). Thereby, the hydrological cycle is an important ingredient in the Earth's climate system due to its transport and redistribution of mass and energy (e.g. Mayewski et al., 2004). To understand the climate during glacial times and to validate climate models proxy data are a prerequisite. Besides proxy data for atmospheric charac-

25 teristics, also the extent and height of the ice sheet must be known, since these have an influence on the atmospheric circulation (Monegato et al., 2017). Still, large uncertainties remain, in particular in land ice-sheet extent and height reconstructions prior to the Last Glacial Maximum (LGM, 21 ka; e.g. Peltier, 1994, 1998; Angelis and Kleman, 2005; Ehlers et al., 2011; Tarasov et al., 2012; Ullman et al., 2014; Batchelor et al., 2019; Gowan et al., 2021), but also the ice-sheet height during the LGM is still debated in literature (e.g. Peltier, 2004; Peltier et al., 2015; Ganopolski and Brovkin, 2017; Batchelor et al., 2019).

Thus, the purpose of this study is to investigate the role of the global and local ice-sheet topography in the regional hydro-climate over the European Alps with a focus on two glacial states, the LGM and the Marine Isotope Stage 4 (MIS4, 65 ka). In this study, we investigate to which extent changes in the dynamics are responsible for precipitation changes over the European Alps, based on eight high-resolution regional climate model (RCM) simulations.

So far, many proxy and global modelling studies have focused on the LGM (e.g. Yokoyama et al., 2000; Clark et al., 35 2009; Van Meerbeeck et al., 2009; Hughes et al., 2013) as the LGM is also a focal period of the Paleoclimate Modelling Intercomparsion Projects (PMIP) (Abe-Ouchi et al., 2015; Kageyama et al., 2017). Globally, the reconstructed temperature at the LGM is reduced by 5 to 6.5 °C compared to present day (PD; Otto-Bliesner et al., 2006). This led to the building up of large ice sheets, in particular over the Northern Hemisphere (Peltier et al., 2015), a strong reduction in the sea level by approximately 115 to 130 m (Lambeck et al., 2014) and changes in vegetation and land surfaces (e.g. Annan and Hargreaves, 40 2013; Bartlein et al., 2011; Cleator et al., 2020), inducing higher atmospheric dust loadings during the LGM (Lambert et al., 2008). Paleoclimate reconstructions of Europe based on pollen data show, depending on the region, a temperature decrease of 10 to 14 °C (Wu et al., 2007; Bartlein et al., 2011). The same data are also used to reconstruct the hydro-climatic response over Europe at the LGM mainly showing drier conditions over Europe (reduction in precipitation of around 200 mm year$^{-1}$) compared to PD (Wu et al., 2007; Bartlein et al., 2011). Roucoux et al. (2005) indicated that the LGM was not necessarily a dry 45 period everywhere in Europe. For instance, Turon et al. (2003), Roucoux et al. (2005), Naughton et al. (2007) and Ludwig et al. (2018) suggested that southern Europe was wetter compared to the rest of Europe and also to adjacent periods, i.e. the Marine Isotope Stage 3 (Voelker et al., 1997; van Kreveld et al., 2000) and the Heinrich events 1 and 2 (Sanchez Goñi and Harrison, 2010; Álvarez-Solas et al., 2011; Stanford et al., 2011). Compared to PD, many studies suggest that the wetter conditions in southern Europe can be explained by a southward shift in the North Atlantic storm track during the LGM (e.g. Hofer et al., 50 2012a; Luetscher et al., 2015; Merz et al., 2015; Ludwig et al., 2016; Wang et al., 2018; Raible et al., 2020; Lofverstrom, 2020). This southward shift is in line with other climate reconstructions that suggest circulation-induced changes in the moisture transport. For instance, Harrison and Digerfeldt (1993) found changes in lake-levels across Europe suggesting major changes in the European atmospheric circulation pattern. Other climate reconstructions also suggest circulation-induced changes in the moisture transport (Florineth and Schlüchter, 2000). In this case, the atmospheric circulation is changed so that the Alpine area 55 receives more moisture from the south which results in wet conditions in the southern part of the Alps and dry conditions north of the Alps. This is confirmed by reconstructions based on speleothems in the Alpine region (Luetscher et al., 2015). Still, the interpretation of sparse paleo-proxy data remains a challenge due to the inherent uncertainties of proxy reconstructions, spatial coverage, uncertain seasonality of the proxy sensitivity, and contradicting signals recorded by different proxy archives (e.g. Wu et al., 2007; de Vernal et al., 2006; Beghin et al., 2016). The MIS4 climate is less understood compared to the LGM

as proxy data availability is further reduced compared to the LGM. Available paleoclimate reconstructions characterise MIS4 to be warmer than the LGM on a global scale (e.g. Eggleston et al., 2016; Newnham et al., 2017; De Deckker et al., 2019) with a global sea level drop of roughly 80 m compared to PD (e.g. Cutler et al., 2003; Siddall et al., 2008, 2010; De Deckker et al., 2019). Focusing on Europe, MIS4 shows relatively wet and warmer conditions at 65 ka compared to the period 85-50 ka (Sánchez Goñi et al., 2013). Still, Sánchez Goñi et al. (2008) and Sánchez Goñi (2020) found drier and colder conditions around the Iberian Peninsula during MIS4 compared to the LGM. Their findings suggest, similar to the LGM period, that the MIS4 climate was not necessarily homogeneously wetter and warmer across Europe.

Global climate model (GCM) simulations offer an alternative view on glacial climate conditions. With respect to the global mean climate response under LGM conditions, they are overall consistent with reconstructions (e.g. Braconnot et al., 2012; Hofer et al., 2012a; Kageyama et al., 2021). On the regional scale, GCM results show stronger deviations to reconstructions, e.g. they tend to underestimate the amplitude of European temperature between LGM and PD or partly disagree in the European precipitation pattern (Braconnot et al., 2012; Kageyama et al., 2017, 2021; Harrison et al., 2015). Besides, GCM simulations are used to deduce relevant processes and assess sensitivity of uncertain components, like the reconstruction of major ice-sheets (e.g. Angelis and Kleman, 2005; Tarasov et al., 2012; Ullman et al., 2014; Peltier et al., 2015; Batchelor et al., 2019). Several studies demonstrated a strong sensitivity of glacial climate to the extent and height of ice sheets (e.g. Kageyama and Valdes, 2000; Rivière et al., 2010; Hofer et al., 2012a, a; Merz et al., 2015). This is particularly true for the Laurentide ice sheet (LIS) as it drives major changes in the glacial atmospheric circulation and its variability compared to PD (Kageyama and Valdes, 2000; Rivière et al., 2010; Pausata et al., 2011; Hofer et al., 2012a; Merz et al., 2015; Harrison et al., 2016). For example, Hofer et al. (2012a) and Merz et al. (2015) found that an increase in the elevation of the LIS causes an enhancement and a southward displacement of the jet stream and the storm track over the North Atlantic. This has a strong impact on the precipitation pattern over Europe, in particular during winter. Still, GCMs operate on relatively coarse resolutions and thus poorly represent the effect of the topography at finer scales such as the complex Alpine terrain. Additionally, GCMs use parameterisations for processes that govern regional-to-local scale precipitation (Leung et al., 2003; Su et al., 2012).

Regional downscaling provides a way to overcome some of the shortcomings of GCM simulations. Latombe et al. (2018) proposed a statistical downscaling method to increase the spatial resolution of LGM simulations in a computationally efficient way. Another approach is dynamical downscaling by employing a regional climate model (RCM; e.g. Strandberg et al., 2011; Rummukainen, 2016; Ludwig et al., 2019). Among others, Ludwig et al. (2016) showed that the application of RCMs substantially improves the simulated LGM climate over Europe compared to the driving GCM, although biases from the GCM simulation may still impact the regional output. For instance, Ludwig et al. (2017) found that RCMs are sensitive to boundary conditions provided by GCMs such as sea surface temperature. Thus, regional climate modelling is beneficial for answering paleoclimate research questions (Ludwig et al., 2019). Pinto and Ludwig (2020) and Raible et al. (2020) showed that extratropical cyclones are characterised by enhanced wind speeds over Europe during the LGM compared to PD, which helps to understand the reallocation and build-up of thick loess deposits in Europe (e.g. Römer et al., 2016; Krauß et al., 2016). Recently, the horizontal resolution of RCMs in the paleoclimate context is increased to a level that convection is explicitly resolved (e.g. Velasquez et al., 2020). This study shows the benefit of higher spatial resolution in particular over areas with

complex terrain such as the Alps. An accompanying study shows that also land surface conditions play an important role and need to be considered to realistically simulate the LGM climate state (Velasquez et al., 2021), which is similar to earlier findings obtained with coarser resolved RCM simulations for LGM and MIS3 (Kjellström et al., 2010; Strandberg et al., 2011; Ludwig et al., 2017). Nevertheless, a detailed analysis of the sensitivity of the global and local ice-sheet topography on the regional hydro-climate over the European Alps — the purpose of this study — is missing. Hence, we employ eight high-resolution RCM simulations with the Weather Research and Forecasting (WRF) model (Skamarock and Klemp, 2008) driven by simulations under constant climate conditions using the Community Climate System Model version 4 (CCSM4, Gent et al., 2011). Thereby, we modify either the height of the northern hemispheric ice sheets, i.e. LIS, the Fennoscandian and Greenland ice-sheet in both CCSM4 and WRF, or the height of the Fennoscandian ice sheets (FIS) in the regional model or solely the height of the Alpine ice sheet in the regional model.

The study is structured as follows. Section 2 describes the models and experiments carried out in this study. Section 3 introduces methods used for analysing the role of the ice-sheet topography on the Alpine hydro-climate. In Sect. 4, we first characterise the two glacial states LGM and MIS4. Secondly, we investigate how the Alpine hydro-climate reacts to changes in the northern-hemispheric ice sheets and the FIS, separately. In a third step, we assess the sensitivity of the Alpine hydro-climate to local (Alpine) ice-sheet changes. Finally, a discussion and conclusive remarks are given in Sect. 5.

## 2 Models and experiments

The study is based on eight experiments using PD and glacial climate conditions to assess the role of the large-scale (LIS and FIS) and local (Alpine) ice-sheet topography on the Alpine climate. The focus on the European Alps necessitates to employ a model chain that consists of a GCM and RCM. Thereby, the GCM provides the initial and boundary conditions for the RCM. The models' configuration and the experiments are explained in the following.

The Community Climate System Model is used as GCM in the model chain (version 4, CCSM4; Gent et al., 2011). We use the atmosphere-land-only setting of CCSM4, i.e. the components of the atmosphere (CAM4, Neale et al., 2010) and land (CLM4, Oleson et al., 2010) are dynamical models whereas the ocean and sea ice components are so-called *data models* obtained from a coarsely resolved fully coupled simulation performed with CCSM3 (Hofer et al., 2012a). Thus, the two data models force the atmospheric component by time-varying sea-surface temperatures and sea-ice cover. The atmosphere-land-only model is run for 33 years with 6-hourly output, a horizontal resolution of $1.25° \times 0.9°$ (longitude $\times$ latitude) and 26 vertical hybrid sigma-pressure levels in the atmosphere and 15 layers in the land component. In this study, we use the first 21 or 12 years of the GCM simulation that follow after the first three years of spin-up. Five global climate simulations provide the RCM with initial and boundary conditions. Two of these CCSM4 simulations are performed under PD 1990 CE conditions and LGM conditions, respectively. The other three simulations are performed under MIS4 conditions. The orbital forcing and atmospheric composition are adjusted to the respective period (Table 1). The MIS4 simulations differ in their northern hemispheric ice-sheet elevation: 66, 100 and 125 % of the LGM ice-sheet elevation, respectively. Note that glaciers and small ice caps such as the ones over the Alps and Pyrenees are not included in these simulations due to the coarse resolution of

approximately 100 km in the GCM. More detailed information on these simulations and their settings are presented in Hofer et al. (2012a, b) and Merz et al. (2013, 2014a, b, 2015).

The global CCSM4 simulations are dynamically downscaled with the RCM Weather Research and Forecasting (WRF) model (version 3.8.1, Skamarock et al., 2008). This RCM solves the basic non-hydrostatic equations with an Eulerian mass-coordinate solver and employs a terrain-following eta-coordinate system in the vertical. We use an adaptive time-step, 40 vertical eta levels and four domains that are two-way nested. The horizontal resolution of these four domains is 54, 18, 6 and 2 km, respectively. The domains focus on the Alpine region; the outermost domain includes Europe and part of the North Atlantic to capture

the influence of the North Atlantic Ocean and the FIS on the European climate (Fig. 1). Note that the domain setup might influence the simulated Alpine climate, which is not possible to test in this study due to the highly expensive model settings. Furthermore, we use the same setting as in Velasquez et al. (2020), thus we refer to this publication on the details of the relevant parameterisation schemes used. Still, we highlight that the horizontal resolutions in the two innermost domains (6 and 2 km) are convection permitting, i.e. we omit the use of parameterisation for convection in these two domains. Also, we use a

relaxation zone of five grid points (e.g. this means 10 km in the innermost domain) at each edge in each domain, which is not included in the analysis. WRF uses 21 or 12 years (compare Table 2 for more details) of the corresponding GCM simulation as initial and boundary conditions, but WRF is not nudged to the GCM output. These 21 and 12 years are further split up into 7 and 4 individual 3-year simulation segments, respectively. We split the simulations to efficiently use the available computer facilities (similar to accompanying studies such as Velasquez et al., 2020, 2021), even though regional climate simulations

would commonly be performed in one single simulation. Note that all segments represent the same climate conditions, i.e. they are driven by the GCM using the same climate state. For each segment, a 2-month spin-up is needed in order to allow the land surface to come into quasi-equilibrium. To determine this spin-up, we analyse each segment and the results show that a 2-month spin-up is sufficient to obtain a quasi-equilibrium of the upper meter of the land surface in this study, i.e. no significant trend in the layers of the WRF land-surface scheme. Accompanying studies also suggest a 2-month spin-up (Velasquez et al.,

2020, 2021). Note that the spin-up might vary if the segment starts in another season, which is not possible to test in this study due to the highly expensive model settings

   The first experiment ($PD_{PD}$) is a reference simulation under PD conditions (1990 CE conditions). We run WRF for 21 years using 1990 CE conditions and initial and boundary conditions of the corresponding CCSM4 simulation (Table 1). $PD_{PD}$ uses the default PD MODIS-based land cover dataset from WRF as land surface boundary conditions (Skamarock et al., 2008).

The second experiment ($LGM_{LGM}$) uses the external forcing of the LGM (Table 1), except for eccentricity and precession. The reason is that in the radiative routine of WRF only the obliquity parameter is processed. We realised this problem after the simulations have been performed. We expect that this problem is of minor importance as the driving CCSM4 uses all orbital parameters and thus the orbital signal is at least partly included in the simulations. Some preliminary results with a model version, which fixes this bug, shows no strong dependence on this error under LGM conditions (Emmanuele Russo pers.

comm.). Additionally, the $LGM_{LGM}$ surface conditions need some further adjustments. These include the lowering of the sea level and ice sheets as specified in the PMIP3 protocol (Fig. 1b; for more details see Ludwig et al., 2017). The glaciation over the Alpine region (obtained from Seguinot et al., 2018) and other glaciated areas (e.g. Pyrenees, from Ehlers et al., 2011) are

modified according to LGM conditions (Fig. 1b). Additionally, the land cover is altered to comply with LGM conditions, as described in Velasquez et al. (2021). Comparing $LGM_{LGM}$ with $PD_{PD}$ illustrates the entire effect of changes in the external

forcing and in the surface conditions (sea level drop, land cover and northern hemispheric ice sheets). Note that the $LGM_{LGM}$ simulation is evaluated against proxy evidence (Prentice and Jolly, 2000; Wu et al., 2007; Kaplan et al., 2016) showing an improved LGM climate state in the WRF simulation (Velasquez et al., 2020, 2021) compared to the driving CCSM4 simulations (Hofer et al., 2012a).

The third to seventh experiments assess the sensitivity of the Alpine climate to changes in the northern hemispheric ice-sheet

configuration. Thereby, the MIS4 simulations of CCSM4 are dynamically downscaled with WRF resulting in $MIS4_{LGM66}$, $MIS4_{LGM}$ and $MIS4_{LGM125}$. These three WRF simulations are run for 21 years using MIS4 conditions and using the $LGM_{LGM}$ land cover (described in Velasquez et al., 2021). We use a $LGM_{LGM}$ land cover because it is the closest approach to a MIS4-like land cover as a gridded European MIS4 land cover has not been developed yet. Even though the $LGM_{LGM}$ land cover might influence the representation of the MIS4 climate, it can provide the opportunity to have some insights into the effects

of a different orbital and atmospheric forcing, i.e. comparing $LGM_{LGM}$ to $MIS4_{LGM}$. Additionally, the Alpine ice sheet is always set to LGM conditions (Fig. 2b). Following their driving CCSM4 simulations, we perform each WRF simulation with a different Fennoscandian ice-sheet thickness: 66, 100 and 125 % of the LGM ice-sheet thickness, respectively. $MIS4_{LGM}$, also written as MIS4 throughout the results and conclusions, serves as reference for $MIS4_{LGM66}$ and $MIS4_{LGM125}$.

To investigate the effect of the FIS on the Alpine climate, we perform two additional sensitivity simulations: $LGM_{FIS50}$ and

$LGM_{FIS150}$. For both simulations, the initial and boundary conditions are from the LGM CCSM4 simulation. Note that this CCSM4 simulation uses 100 % LGM ice-sheet configurations for both sensitive simulations. To assess the influence of the FIS, we reduce or increase FIS thickness to 50 % and 150 %, but only in the WRF model. Both simulations are run for 12 years.

In the experiment number eight, we investigate the influence of local modifications of the Alpine ice sheet on the Alpine climate. One sensitivity simulation is performed: $LGM_{ALPSLESS}$. This simulation is based on the $LGM_{LGM}$ (which serves

as reference) but with a modified Alpine topography (Fig. 2c). $LGM_{ALPSLESS}$ is run for 21 years and with a reduced Alpine glacier thickness. This reduction becomes stronger with height; namely, the Alpine ice sheet is strongly reduced over mountain peaks and slightly over the low lands. All eight experiments are summarised in Table 2.

## 3  Methods

The analysis of the past hydro-climate and its response to different forcing factors is based on climatological means of precipi-

tation, their differences between the experiments and some of the driving factors for these differences, e.g. changes in humidity or wind. We assess the statistical significance with a bootstrapping technique (Wilks, 2011). This technique randomly selects elements from the original sample to generate a new sample, also called resampling, whereby the number of elements remains unchanged. This procedure is repeated 1000 times. A new mean value is calculated from each resampling leading to 1000 mean values that are used to build a probabilistic distribution function (PDF). We assess the significance of the mean value

using a significance level of 0.05 (0.025 for each PDF's tail). The bootstrapping technique is applied at each grid point using as

elements the annual mean values. Note that few results may be considered negligible, e.g. due to their very small differences, even though there is a statistical significance given by the bootstrapping technique.

Additionally, we analyse vertical atmospheric profiles using the SkewT–Log P diagram (AWC, 1969, 1990; NOAA, 2021) to gain insights into the atmospheric drivers of precipitation changes. The SkewT–LogP or Skew–T diagram is a thermodynamic
diagram that is widely used in atmospheric science, particularly by meteorologists for weather forecasts (e.g. Duarte and Gomes, 2017; Morsy et al., 2017; da Silva et al., 2019; Chen et al., 2020). It illustrates the vertical atmospheric state by several meteorological variables such as air temperature, humidity, wind speed and wind direction. The $x$-axis indicates temperature and the $y$-axis pressure levels in a logarithmic scale (thus the name LogP). There are usually five isolines on the diagram: isotherms, isohumes (or saturated mixing ratio lines), dry adiabatic lines (or lines of equal potential temperature $\theta$), isobars,
and saturation adiabatic lines (also known as moist adiabatic lines). In this study, we use a simplified diagram that only contains the first three variables for the analysis as we focus on changes in temperature, humidity and wind. For instance, an unsaturated air parcel follows a dry adiabatic line when ascending without changes of state, i.e. there is no loss or gain of latent heat. These lines are used in this study to qualitatively estimate the stability of the atmosphere. A stable atmosphere is characterised by an increase of potential temperature with height.
Two vertical profiles are included in the Skew–T diagram: air temperatures (solid lines) and dew-point temperatures (dashed lines). The air temperatures are used to deduce the mixing ratios with height. The real mixing ratios with height, i.e. the amount of water vapour, are obtained from the saturated mixing ratio lines when they are crossed by the dew-point temperature vertical profile. The dew-point temperatures indicate the temperatures at which the air becomes saturated. Both air and dew-point temperatures are used to investigate the relative humidity, i.e. the level of saturation at a certain pressure and temperature. Note
the Skew–T diagram implicitly includes the dependency of the relative humidity on temperature by the logarithmic scale of the saturated mixing ratio lines. Therefore, the relative humidity can be obtained by qualitatively estimating the distance between both temperature profiles. A short distance indicates a high relative humidity and, inversely, a large distance a low relative humidity. Wind speed and direction are illustrated by wind bars in km h$^{-1}$.

The Skew–T diagram is built with climatological means of the atmospheric variables above surface at the following pressure
levels: 1000, 925, 900, 850, 800, 750, 700, 600, 500, 400, 300, 250, 200 and 100 hPa. We consider the lowest pressure level above surface as the best representation of the near-surface atmosphere. Two additional vertically integrated quantities are given at the top of the diagram: precipitable water (PW) and convective available potential energy (CAPE). PW represents the amount of water vapour of an atmospheric column expressed as the depth of water if that vapour were condensed, i.e. water vapour available for precipitation. The PW value is calculated by vertically integrating the water vapour mixing ratios across
height and it does not depend on the temperature (e.g. Rozsa, 2012; González-Rojí et al., 2018). CAPE quantitatively represents the energy available for convective processes, the higher it is the stronger these processes could be (for more information see chapter 8 of Wallace and Hobbs, 2006). In this study, we use the climatological mean values to calculate the PW and CAPE values; the later are only shown at the top of the diagram when it is different from zero.

## 4 Results

In the following, the glacial Alpine hydro-climate is characterised for two glacial states: the LGM and MIS4. Then, we investigate the sensitivity of the hydro-climate to northern hemispheric ice-sheet changes for MIS4 and the sensitivity to the FIS for LGM. We assess the Alpine hydro-climate response to changes in the Alpine ice sheet by one simulation using LGM conditions. For the analysis, we select winter (DJF, December-January-February) and summer (JJA, June-July-August) as these two seasons summarise the changes in spatial precipitation patterns over the Alpine region. The other two transition seasons (MAM, March-April-May and SON, September-October-November) show either similarities to winter (for MAM) or summer (for SON). The analysis is based on the innermost domain of the WRF simulations (Fig. 2).

### 4.1 Characterisation of the glacial hydro-climate of the Alps

Present-day hydro-climate over the Alpine area is characterised by the large scale atmospheric circulation with its dominant westerlies, influences from the Mediterranean (Messmer et al., 2015, 2017, 2020) and convective processes (Ban et al., 2014; Gómez-Navarro et al., 2018). In winter, the westerlies are enhanced, so that we observe high precipitation amounts in the north-western area of the domain and at the northern face of the Alps, whereas the southern and the eastern parts receive less precipitation (Fig. 3a). Due to the orographic barrier of the Alps the flow is uplifted leading to a higher precipitation intensity over mountain tops and lower intensity in valleys. In summer, the precipitation pattern is more uniform over the Alps, showing high intensities also on the southern face of the Alps and in the east (Fig. 3e). This suggests that the topographically triggered convection is an important process in summer under PD climate.

The LGM shows a completely different precipitation behaviour over the Alps (Fig. 3). In winter, precipitation is mostly concentrated on the south-western Alps (Fig. 3b and c) with a statistically significant increase of more than 8 mm day$^{-1}$ compared to PD (Fig. 3c). The southern face of the Alps generally shows a statistically significant increase of winter precipitation of about 3 mm day$^{-1}$ during LGM compared to PD, whereas the northern face experiences a statistically significant decrease of more than 6 mm day$^{-1}$ (Fig. 3c). During summer, precipitation is significantly reduced at the LGM compared to PD (Fig. 3e, f and g), which suggests that convective processes, typically observed in PD summer, are less active during the LGM. Still, few areas show wetter conditions during the LGM compared to PD, particularly over mountains peaks in the western Alps (Fig. 3g). The increase in precipitation observed over mountain peaks indicates precipitation induced by orographic lifting. As the surroundings are drier it is assumed that the higher elevation of the Alps during LGM triggers precipitation only at the highest points.

To further understand the precipitation changes between LGM and PD, we use the Skew–T diagram, as introduced in Sect. 3. The vertical profiles are estimated for two sites: upstream to the Alpine region (site A, north-western region) and downstream (site B, central-southern region) according to the general westerlies of the mid latitudes (Fig. 2). Note that these two sites are over rather flat terrain and therefore experience less influence from local orographic-induced atmospheric dynamics such as the mountain-valley breeze.

Starting with the Skew–T diagrams in winter (Fig. 4a and c), we find the expected strong reduction of LGM temperatures compared to PD at both sites. The mixing ratios (the saturated mixing ratio lines crossed by the dew-point temperature vertical profile) are also reduced in the LGM compared to PD resulting in less precipitable water under LGM conditions at both sites. Interestingly, the relative humidity, measured by the distance between dew-point temperature and temperature remains un-

changed when comparing LGM and PD. However, we find a clear structural change in the vertical profiles of the temperatures, as illustrated by the different slopes between LGM and PD. Both sites show that potential temperature values (comparing the temperature profile with the dashed brown lines) overall increase with height indicating a stable atmosphere. This increase is stronger in the LGM indicating a more stable atmosphere at both sites compared to PD. Particularly, the stability is higher in site A (north-western part) than in site B (central-southern region) in winter, which is a first hint that both regions show differ-

ent behaviours in their winter precipitation. Another hint is given by the wind speed and direction, since we find an increase in wind speed and an anticlockwise turning of the winds during the LGM compared to PD. This is evident at both sites but the turning of the winds is more pronounced in the central-southern region. The boundary layer is thicker in the LGM than in PD, in particular for the central-southern region. Both sites show the expected lowering of the tropopause during the LGM due to colder conditions compared to PD.

Summer shows a rather similar behaviour as winter, but with a few exceptions related with the season (Fig. 4b and d). CAPE is only observed during PD summer at both sites. Additionally, we find a development of a small boundary layer under PD and LGM conditions, particularly distinct during the LGM. Furthermore, the wind is turned slightly clockwise at site A (north-western region), whereas winds turn anticlockwise at site B (central-southern region). Again, at both sites, the wind speed is increased in the LGM, but not as clearly as in winter.

To gain further insights in the advection of moisture, we first focus on the larger-scale mid-atmospheric circulation for the second domain (over Europe at 18 km resolution). The comparison to PD conditions shows that the climatological mid-atmospheric mean flow turns anticlockwise with stronger wind speeds, which indicates a more zonal, intensified and southward shifted winter jet stream during the LGM. Same results were already found in previous studies over Europe, which analysed the CCSM4 simulations that drive our WRF simulations (Hofer et al., 2012a, b). This indicates that there is a strong correlation of

the larger-atmospheric circulation between the driving CCSM4 and the WRF simulations. Note that these CCSM4 simulations and their underlying atmospheric circulation have been already analysed over Europe in a variety of studies (e.g. Hofer et al., 2012a, b; Merz et al., 2013, 2014b, a, 2015, 2016; Landais et al., 2016); therefore please refer to these studies for a in-detail analysis of the European atmospheric circulation. Secondly, we exhibit the wind vectors at 700 hPa for the innermost domain (over the Alpine region at 2 km resolution). The winds at 700 hPa summarises atmospheric circulation in the low-to-mid

troposphere over the Alpine region at a higher spatial resolution. Note that this pressure level (700 hPa) could also represent near-surface winds over mountain peaks in certain regions. Compared to PD, winds are significantly stronger and turned anticlockwise in winter during the LGM (Fig. 5a), confirming the finding of the Skew–T diagrams (Fig. 4 a and c). Almost the entire innermost domain shows significant changes in both wind components (red shading in Fig. 5a). Stronger winds are also observed during summer but with slightly different turning patterns across the innermost domain (Fig. 5c). These winds are

generally turned anticlockwise except for the southwestern region of the innermost domain where it is clockwise. The increase

in speed covers approximately half of the innermost domain, which is attributed to either significant changes in the zonal component (U) only or both wind components (i.e. blue or red shading, respectively; Fig. 5c). Similarly, the wind turning also covers about half of the innermost domain, which is associated with either significant changes in the meridional component (V) only or both wind components (i.e. areas in grey or red, respectively; Fig. 5c).

MIS4 is the second glacial state considered here. Figure 3d shows that winter precipitation intensities of MIS4$_{LGM}$ are higher than LGM ones, especially over some areas such as the western area of the domain with a significant increase of about 3 mm day$^{-1}$. This is expected as MIS4$_{LGM}$ climate is warmer than LGM, thus the ability of the atmosphere to hold more moisture is increased due to the Clausius–Clapeyron equation (e.g. third chapter of Wallace and Hobbs, 2006). In summer, the precipitation difference between MIS4 and LGM shows a significant north-south dipole pattern (Fig. 3h). There are slightly
wetter conditions of about 1.5 mm day$^{-1}$ on the northern face of the Alps during MIS4$_{LGM}$ compared to LGM. On the southern face of the Alps precipitation is reduced by around 2 mm day$^{-1}$.

To understand these differences, we again investigate the atmospheric vertical profiles at both sites. Overall, the MIS4 profiles look very similar to LGM ones during winter. The expected shift towards warmer conditions under MIS4 conditions is visible at both sites (Fig. 4a and c), in particular in the lower part of the troposphere (mainly in the boundary layer), leading
to higher mixing ratios (saturated mixing ratio lines crossed by the dew-point temperature vertical profile) in MIS4 compared to LGM. The comparison of MIS4 and LGM further shows that the stability is slightly reduced under MIS4 conditions in the lower part of the troposphere (up to 600 hPa; Fig. 4a). This reduction is more evident during winter in both regions, i.e. north-western (site A, Fig. 4a) and central-southern (site B, Fig. 4c). The wind directions seem to be unchanged with slightly lower wind speeds under MIS4 than LGM conditions. Thus, thermodynamic changes are the major processes in explaining the
increased winter precipitation during MIS4 (Fig. 3d). In summer, the temperature profiles agree between the two glacial states at both sites, except for a shift towards higher temperatures leading again to higher mixing ratios. The only deviation is that the boundary layer shows a slightly stronger inversion during MIS4 than LGM at both sites. The shift to warmer temperature and higher mixing ratios suggest a general moistening under MIS4 compared to LGM conditions, which can explain the increase in summer precipitation in the northern part but not the decrease in the southern part of the domain.

Therefore, we assess the wind vectors at 700 hPa for the innermost domain (over the Alpine region at 2 km resolution). Note that the LGM and MIS4$_{LGM}$ simulations use the same topography, i.e. the LGM topography. Figure 5b shows that MIS4 winds become weaker during winter compared to LGM, but a turning is almost absent. During summer, MIS4 winds become stronger with a slight clockwise turning compared to LGM (Fig. 5d). Both, the increase in speed and the turning, enhances the wind shear in the low-to-mid troposphere (Fig. 4) resulting in more convective-related precipitation in the northern part of the
Alps. Over the southern face of the Alps, the slightly clockwise turning could lead to reduced moisture availability in MIS4 as the flow would tend to dry out when crossing the Alps and reaching the Po valley (assuming a Foehn process)

In summary, we find that both thermodynamic and dynamical changes are responsible to generate precipitation changes in winter and summer when comparing LGM and PD conditions and MIS4 and LGM, respectively. Interestingly, the changes in winter precipitation between LGM and PD is explained by a combination of thermodynamic and dynamic processes, whereas

the summer reduction is mainly explained by thermodynamics and reduced convection. The comparison of MIS4 and LGM shows that the winter changes are mainly driven by thermodynamics, whereas in summer also dynamical changes are important.

## 4.2 Sensitivity of the Alpine hydro-climate to northern hemispheric ice-sheet changes

Here, we investigate the role of the northern hemispheric ice-sheet topography in the Alpine climate. Two sets of sensitivity simulations are used. The uncertainty in the thickness of the northern hemispheric ice sheets is assessed by comparing two
simulations with 66 and 125 % to the one with 100 % LGM ice-sheet thickness under MIS4 conditions. The second set of simulations uses the LGM climate state as a reference and only the FIS thickness is changed by 50 and 150 %.

We first focus on the precipitation response to these changes in ice-sheet thickness. The comparison of $MIS4_{LGM66}$ with $MIS4_{LGM}$ shows that lowering the northern hemispheric ice sheets by 66 % significantly increases the winter precipitation by about 3 mm day$^{-1}$ on the northern face of the Alps and leads to significantly drier conditions in the rest of the domain (Fig.
6b). In particular, winter precipitation is reduced by up to 8 mm day$^{-1}$ in the south-western Alps (Fig. 6b). Comparing these patterns to the difference between PD and LGM, we find a north-south winter pattern inversely to the one of PD and LGM. This suggests that decreasing ice-sheet thickness (i.e. $MIS4_{LGM66}$) leads to more PD-like conditions during glacial climates. Summer precipitation differences between the two experiments resembles the winter pattern, but the amplitudes are reduced, i.e. precipitation is increased by about 2 mm day$^{-1}$ on the northern face of the Alps and reduced by about 1 mm day$^{-1}$ in
the western part of the domain (Fig. 6e). The southern part shows no significant changes. This pattern of precipitation is only partly similar to the difference pattern of the LGM and PD.

An increased ice-sheet thickness as in $MIS4_{LGM125}$ shows a different influence on precipitation patterns (Fig. 6c and f). In winter, the difference pattern in precipitation between $MIS4_{LGM125}$ and $MIS4_{LGM}$ is generally similar to the one found between LGM and PD. Especially, we find significantly high precipitation intensities up to 3 mm day$^{-1}$ on the north western
and southern regions of the domain (Fig. 6c).. The northern face of the Alps shows a decrease in the precipitation intensities. Thus, we interpret that the response of winter precipitation is linear with respect to the northern hemispheric ice-sheet thickness changes. In summer, we generally find drier conditions in the $MIS4_{LGM125}$ than in $MIS4_{LGM}$, in particular significantly lower precipitation of up to -3 mm day$^{-1}$ on the central to southern part of the domain (Fig. 6f). This indicates that increasing northern hemispheric ice-sheet thickness (such as in $MIS4_{LGM125}$) enhances glacial climate conditions.
Secondly, we focus on the other set of sensitivity simulations under LGM conditions, where only the FIS thickness is varied. These simulations show a rather weak response of the precipitation within the domain (Fig. 7). In winter, increasing or decreasing the FIS thickness does not lead to significant changes in precipitation (Fig. 7a and b). In summer, we find an increase in the southern part of the domain for both sensitivity simulations (Fig. 7c and d) and a small but significant increase in the northeast of the domain in the $LGM_{FIS150}$ simulation.
Again, the Skew–T diagram is used to understand the role of the vertical behaviour of the atmosphere on precipitation changes at the two sites (Fig. 2). The sensitivity simulations of the northern hemispheric ice-sheet height shows that at both sites the wind speed in winter is enhanced and turned anticlockwise with increasing ice-sheet thickness (Fig. 8a and c). While decreasing northern hemispheric ice-sheet thickness leads to slightly weaker and clockwise turned winds and a slightly warmer

atmosphere. Even though the Skew-T diagram indicates that the relative humidity is rather low, the warmer atmosphere results
in a small increase of moisture availability in the middle-to-low atmosphere (water vapour) in winter. Note that the higher
moisture availability is illustrated by the increase of the values of the mixing ratio, which are obtained from crossing the
saturated mixing ratio lines with the vertical profile of the dew-point temperature. This slight moisture increase is found in the
central-southern region (site B, Fig. 8c) where there is more precipitable water (PW values at the top of Fig. 8c). In summer,
only small wind changes are found at both sites (Fig. 8b and d) and changes in the other variables of the Skew–T diagram are
mainly restricted to the lower troposphere and the boundary layer (Fig. 8b and d). We find a reduced relative humidity at site A
(north-western region), measured by the distance between the dew-point temperature and temperature, and lower mixing ratios
(saturated mixing ratio lines crossed by the dew-point temperature vertical profile) at site B (central-southern region) with
increasing ice-sheet thickness. This suggests dryer conditions at both sites in the case of $MIS4_{LGM125}$. The Skew–T diagrams
for the sensitivity of the FIS do not show strong differences (therefore not shown). Thus, the Skew–T analysis confirms that
an increase in northern hemispheric ice-sheet thickness enhances glacial climate conditions in the Alpine region. It further
suggests that similar processes as discussed in the comparison between LGM and PD are responsible for precipitation changes,
whereas only changing the FIS does not seem to have strong implications.

The Skew–T diagrams already give some hints to wind alterations with respect to changes in the northern hemispheric ice-
sheet topography. Thus, we further assess the role of the northern hemispheric ice-sheet topography on the Alpine winds by
showing the wind vectors at 700 hPa for the innermost domain (over the Alpine region at 2 km resolution). Overall, we observe
that the 700-hPa wind speed is weaker with decreasing ice-sheet thickness. In winter, we find that winds turn clockwise over
the entire domain with decreasing northern hemispheric ice-sheet thickness (Fig. 9a and b). These modifications are generally
associated with either significant changes in the meridional component (V) only or both wind components (i.e. grey or red
shading, respectively; Fig. 9a and b). In summer, we find that winds turn clockwise with decreasing ice-sheet height (Fig. 9c
and d). The alterations in summer are related to either significant changes in the zonal and meridional component (U and V)
only or both wind components (i.e. blue, grey or red shading, respectively; Fig. 9c and d). Thus, the winter and summer wind
fields react similar to the comparison between LGM and PD suggesting that northern hemispheric ice-sheet thickness is an
important driver of the advection processes over the Alps. Again, the sensitivity simulations with changed FIS thickness do not
show significant changes in the wind field (therefore not shown), which again confirms the findings that the thickness of the
FIS seems to have limited influence on Alpine precipitation.

In summary, the analysis shows that winter and summer seasons react differently to northern hemispheric ice-sheet thickness
changes, but resemble the processes already found in the comparison between LGM and PD: Increasing northern hemispheric
ice-sheet thickness generally leads to enhanced glacial conditions, i.e. a moistening during winter due to dynamic processes
and a drying in summer mainly explained by thermodynamics. The sensitivity of precipitation to the FIS thickness is rather
weak and the simulations suggest that its thickness has only a negligible influence on the Alpine precipitation.

### 4.3 Sensitivity of the Alpine hydro-climate to Alpine ice-sheet changes

Besides changes in the northern hemispheric ice sheet, modifications in the Alpine ice sheet might also influence Alpine precipitation. Therefore, we investigate the precipitation pattern of the sensitivity simulation $LGM_{ALPSLESS}$ in comparison to $LGM_{LGM}$. Then, we assess the processes explaining these changes using again the Skew–T diagram and the wind field at 700 hPa.

In winter, the $LGM_{ALPSLESS}$ experiment shows some areas with significantly drier conditions over the Alps compared to $LGM_{LGM}$ (Fig. 10a), especially in the western Alps with some significant precipitation reductions of up to 6 mm day$^{-1}$. This dryness generally coincides with the reduced Alpine ice-sheet thickness (Fig. 2c). Interestingly, the reduction in precipitation is higher in the western part than in the central to eastern part of the Alps, although the Alpine ice-sheet thickness is reduced more strongly in the east than in the west (Fig. 2c). In summer, changes of the Alpine ice sheet go along with a significant north-south precipitation pattern (Fig. 10b). Most of the significant precipitation changes are found in the central to eastern Alps, whereas only a small reduction in precipitation is evident in the western part. Thus, this modification in summer follows the west-east gradient of the Alpine ice-sheet thickness reduction (Fig. 2c). Overall, both seasons demonstrate that locally heterogeneous changes in the topography significantly influence local precipitation patterns over the Alps.

To further understand these precipitation changes, we investigate the vertical profiles at the two sites (Fig. 2) and the wind field at 700 hPa. The Skew–T diagrams of both sites and both seasons show no structural change in the temperature, moisture, and thus the stability of the atmosphere when comparing the $LGM_{ALPSLESS}$ with the $LGM_{LGM}$ simulation (therefore not shown). This is somehow expected, since the two sites are located in areas where the Alpine ice sheet is not present. Nevertheless, we interpret this result that precipitation changes are restricted to the area where the ice sheet is changed, i.e. no downstream effects are found in the temperature and moisture profiles and thus in the stability. This is in contrast to wind, where some changes are evident at both sites. These changes are better illustrated in the wind field at 700 hPa (Fig. 11). In winter, the wind field of the $LGM_{ALPSLESS}$ experiment only changes significantly over the central to western Alps and in some areas south of the Alps (Fig. 11a). The $LGM_{ALPSLESS}$ winds are a bit stronger and slightly turned clockwise compared to $LGM_{LGM}$. The slight turning is associated with significant changes in V at the northern face of the Alps, in both wind components over the Alpine axis and in U in the south of the Alps (Fig. 11a). During summer, we overall observe a similar behaviour as in winter but with an eastward extension of the changes (Fig. 11b). Note that wind changes in winter cannot directly explain the stronger reduction of winter precipitation in the western part (see previous paragraph), which are caused by the reduction of the Alpine ice-sheet thickness. The reason is that we would expect wetter conditions in the western part since the winds slightly turn clockwise (more perpendicular to the Alps) and slightly stronger. Additionally, the further analysis of the larger-scale shows no differences (therefore not shown).

In summary, precipitation changes in both seasons are associated with the fact that the wind field faces a lower orographic barrier due to a reduction of the Alpine ice sheet. This effect results in reduced (increased) precipitation at the northern face of the Alps in winter (summer). Additionally, this leads to drying the south-western Alps in winter and the southern face of the Alps in summer.

## 5 Discussion and conclusions

In this study, we investigate the sensitivity of the glacial Alpine hydro-climate to northern hemispheric and local ice-sheet changes. To that end, we employ a GCM-RCM model chain to perform sensitivity simulations for two glacial periods, the LGM and MIS4. The LGM is compared to the PD and MIS4 climate simulation in order to characterise these glacial states. Then, we assess the impact of northern hemispheric ice-sheet thickness on the Alpine hydro-climate under MIS4 and LGM

conditions. The second sensitivity test uses LGM conditions as base line and assesses the hydro-climatic response to changes in the Alpine ice sheet. Note that the results might depend on the model setup, which is not possible to test in this study due to the highly expensive model settings.

The LGM is known to be a period of generally drier and colder conditions than today (Otto-Bliesner et al., 2006). Earlier studies using the same LGM WRF simulation (e.g. Velasquez et al., 2020, 2021) showed that an application of the GCM-RCM

model chain is beneficial with respect to temperature and precipitation over Europe compared to the driving GCM (Hofer et al., 2012a, b; Merz et al., 2013, 2014a, b, 2015, 2016) and other global model simulations (e.g. Kageyama et al., 2017, 2021), since the last ones underestimate the temperature amplitude between PD and LGM.

Here, we focus the analysis on the hydro-climate over the Alps. In winter, we find wetter conditions in the southern part of the Alps during LGM compared to PD (Frei and Schär, 1998; Schwarb et al., 2001). The northern part, however, is dryer under

445 LGM conditions, which is expected due to the general colder conditions of the LGM (as the Clausius–Clapeyron equation suggests). This enhanced north-south precipitation gradient resembles finding of Becker et al. (2016) showing that such a gradient is a prerequisite to explain the extent of the Alpine glacier during the LGM. Even though LGM climate was colder with lower mixing ratios (saturated mixing ratio lines crossed by the dew-point temperature vertical profile), our analysis shows that changes in the wind speed and direction substantially contribute to the north-south precipitation pattern. Winds

are significantly stronger and anticlockwise turned over the Alpine region during LGM suggesting an increase of intensity or frequency of the moisture advection from the south to the Alps. This is in line with proxy evidence (Florineth and Schlüchter, 2000; Luetscher et al., 2015; Spötl et al., 2021). These authors similarly found a circulation change from dominant westerlies during PD to a more southern atmospheric circulation during the LGM. To explain these changes, global modelling studies suggested a southward shift of the North Atlantic storm track during the LGM compared to PD (e.g. Hofer et al., 2012a;

Luetscher et al., 2015; Merz et al., 2015; Raible et al., 2020) and a change in the weather patterns (e.g. Hofer et al., 2012b; Ludwig et al., 2016; Wang et al., 2018). Thus, our analysis shows that changes in the atmospheric dynamics on the regional to local scale are also relevant to explain precipitation changes, in particular the moistening of the southern face of the Alps. During summer, we find drier conditions in most of the Alpine domain. This is expected, as the LGM is generally colder than PD (Clausius–Clapeyron equation). Additionally, we find a strong reduction in convective activity, which can be traced back

to a colder atmospheric state and an increase in stability during the LGM compared to PD.

The MIS4$_{LGM}$ climate shows enhanced winter precipitation compared to the LGM. The reason is that the MIS4 climate state is generally warmer (Hofer et al., 2012a, b; Merz et al., 2013, 2014a, b, 2015, 2016) and thus more moisture is globally available. Wind changes do not contribute to these wetter conditions in the Alpine region as they become weaker and therefore

reduce the orographically forced uplifts, which also suggests an overall reduction of the moisture transport to the Alps. Thus, we interpret the winter changes between MIS4 and LGM to be purely thermodynamically driven (Clausius–Clapeyron equation). In summer, $MIS4_{LGM}$ shows slightly wetter conditions at the northern face and drier conditions at the southern side of the Alps. The northern wetter conditions are induced by an increase in the tropospheric vertical wind shear enhancing convection processes. The drier conditions at the southern face of the Alps may be explained by an enhanced Foehn effect due to the slightly clockwise turned winds (statistically significant). Thus, dynamical processes could also play a role to explain the summer precipitation changes between MIS4 and LGM.

The northern hemispheric ice-sheet topography strongly influences the precipitation over the Alpine region. In both seasons, the precipitation patterns and the related thermodynamic and dynamic processes are similar to the ones found in the comparison between the LGM and PD. Namely, an increase of the northern hemispheric ice sheet leads to an intensification of glacial conditions over the Alps. Changes in the FIS do only weakly alter the precipitation patterns over the Alpine region. One potential reason of this weak precipitation response may also be the design of the Fennoscandian sensitivity experiment as the driving GCM has not experienced the changes of the FIS, as the FIS was only modified in the RCM. However, we introduced rather strong changes in the RCM; thus, we expect only a minor impact of the experimental design on the conclusion that the FIS is less influential on the Alpine precipitation. We further conclude that changes in the Laurentide ice sheet needs to be considered in the estimation of Alpine precipitation. Moreover, the analysis shows that the northern hemispheric ice-sheet thickness is mainly responsible for the dynamical processes explaining the precipitation changes. This is suggested by the similarity of the processes found in the sensitivity experiments of northern hemispheric ice-sheet thickness and in the comparison between the LGM and PD.

In the assessment of the role of the Alpine ice-sheet topography, we found significant changes mainly over the area where the ice sheet was altered and south to this area, e.g. south to the Alps during summer. These changes are not as strong as the ones identified for changes in the northern hemispheric ice sheet or between LGM and PD. Nevertheless, they are responsible for a redistribution of precipitation over the Alps, e.g. a stronger reduction in the western part than in the central and eastern part during winter. These changes are relevant for glacier modelling (e.g. Jouvet et al., 2017; Seguinot et al., 2018). Thus, the analysis presented here suggests that future modelling efforts should ideally involve coupled glacier regional climate models. At the moment, this is not possible due to the long calculation time needed for glacier models and the high computational cost of RCMs. An intermediate step is to use the output of different sensitivity simulations, as presented here, in ice-sheet modelling studies (e.g. Jouvet et al., 2017; Seguinot et al., 2018).

Moreover, future studies will benefit from even more detailed climate simulations over the Alpine region, particularly to better understand precipitation patterns in complex terrain. Both, the climate variables but also a better understanding of the ice-sheet dynamics would be beneficial for studies on the local and regional paleobotany (Kaplan et al., 2016), archaeology (Burke et al., 2017; Wren and Burke, 2019) and anthropology (e.g. Maier et al., 2016). For instance, future studies will be able to perform a back-trajectory analysis to identify the moisture source that contributes to the precipitation changes over the Alps. Still, using a larger innermost domain would be beneficial in a future work to enhance the model-related development of the atmospheric circulation around the Alps. Additionally, since the results of this study may depend on the chosen global

and regional climate model, future modelling efforts are needed to perform more regional paleoclimate simulations, especially
using different models to develop a model ensemble. This ensemble would allow to better assess the uncertainties of the
simulated glacial climates

*Code and data availability.* WRF is a community model that can be downloaded from its web page (https://www2.mmm.ucar.edu/wrf/users/,
last access 04 April 2022; Skamarock and Klemp, 2008). The climate simulations (global: CCSM4 and regional: WRF) and land cover sim-
ulations (LPJ-LMfire) occupy several terabytes and thus are not freely available. Nevertheless, they can be accessed upon request to the con-
tributing authors. Simple calculations carried out at a grid point level are performed with Climate Data Operator (CDO, Schulzweida, 2019)
and NCAR Command Language (NCL, UCAR/NCAR/CISL/TDD, 2019). The figures are performed with NCL (UCAR/NCAR/CISL/TDD,
2019).

*Author contributions.* PV and CCR contributed to the design of the experiments. PV carried out the climate simulations and wrote the first
draft. M.M. provided support in the initialisation of WRF and in the performance of the base-line simulations (PD and LGM). All authors
contributed to the interpretation of the results, the writing, and scientific discussion.

*Competing interests.* The authors declare no competing interests.

*Acknowledgements.* This work was supported by the Swiss National Science Foundation (SNF) within the project 'Modelling the ice flow
in the western Alps during the last glacial cycle'. MM is supported by the SNF Early Postdoc.Mobility programme. The simulations are per-
formed on the super computing architecture of the Swiss National Supercomputing Centre (CSCS). Data is locally stored on the oschgerstore
provided by the Oeschger Center for Climate Change Research (OCCR).

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

**Table 1.** External forcing used in this study for 1990 CE, LGM and MIS4 conditions. The orbital parameters are calculated according to Berger (1978). Estimates for glacial levels of $CO_2$, $CH_4$ and $N_2O$ are obtained for LGM from the PMIP protocol (http://www-lsce.cea.fr/pmip2/) and for MIS4 from ice cores according to Schilt et al. (2010) and Bereiter et al. (2014). Note that the external forcing corresponds to the values of the driving the CCSM4 simulations (Hofer et al., 2012a, b)

| Parameter name | 1990 CE | LGM | MIS4 |
| --- | --- | --- | --- |
| TSI (W m$^{-2}$) | 1361.77 | 1360.89 | 1360.89 |
| Eccentricity ($10^{-2}$) | 1.6708 | 1.8994 | 2.0713 |
| Obliquity (°) | 23.441 | 22.949 | 22.564 |
| Angular precession (°) | 102.72 | 114.43 | 195.22 |
| $CO_2$ (ppm) | 353.9 | 185 | 205 |
| $CH_4$ (ppb) | 1693.6 | 350 | 460 |
| $N_2O$ (ppb) | 310.1 | 200 | 210 |

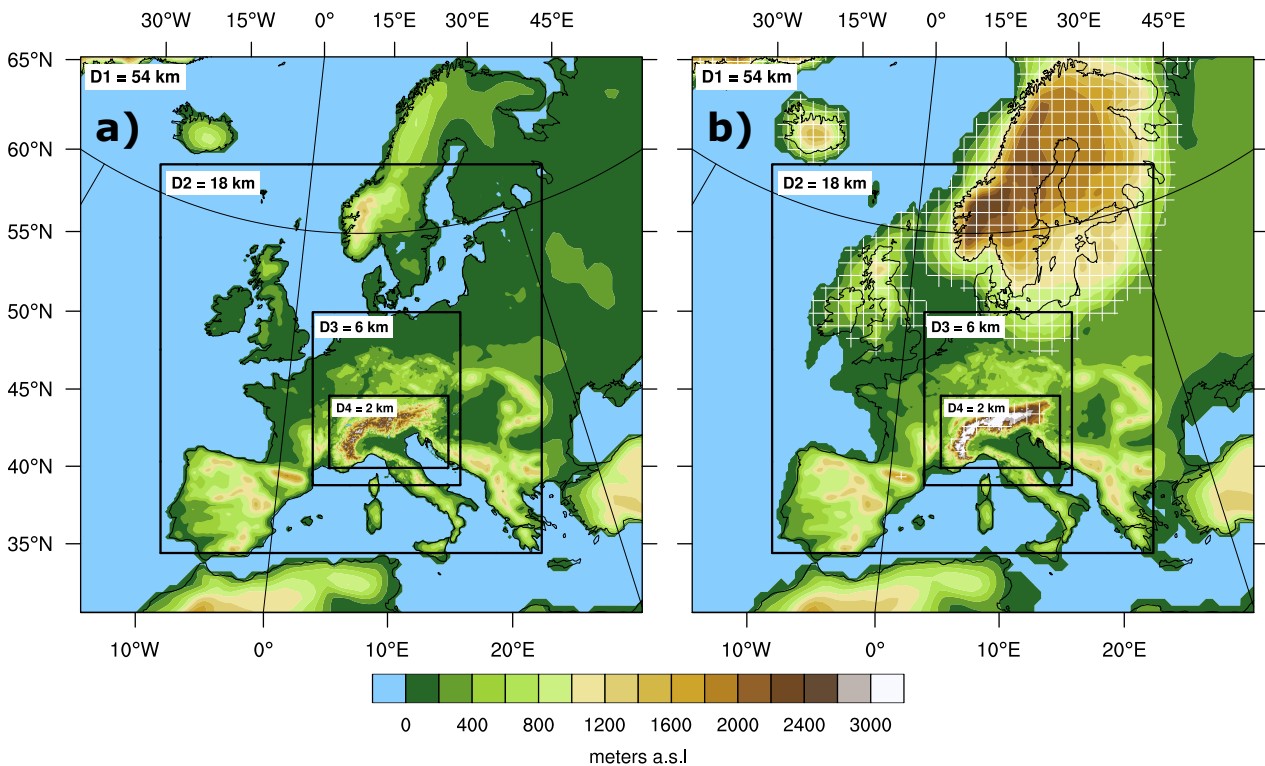

**Figure 1.** Domains and topography used by WRF. (a) represents the four domains at 54, 18, 6 and 2 km horizontal resolution and the shading indicates present-day topography, (b) as (a) but for the LGM topography, crosshatched areas are covered by glaciers.

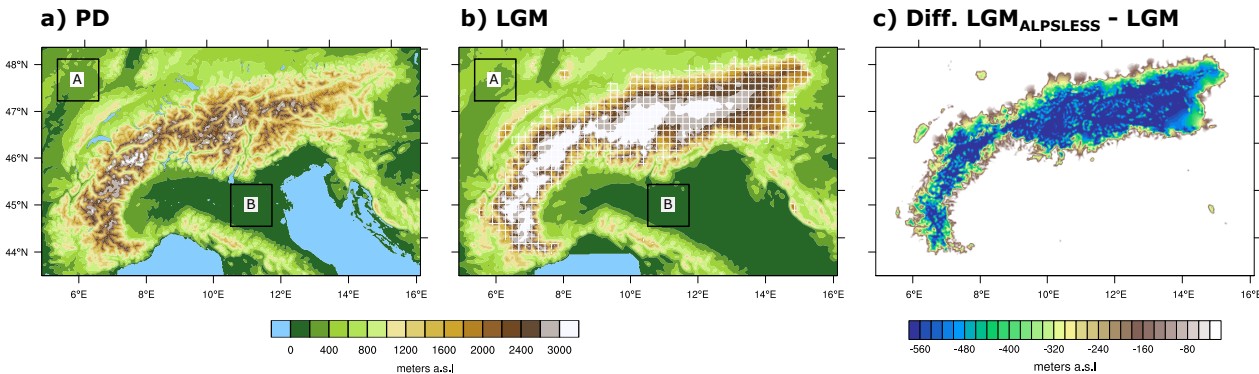

**Figure 2.** Innermost domain and topography used by WRF. (a) represents the domain at 2 km horizontal resolution and the shading indicates PD topography, (b) as (a) but for the LGM topography, crosshatched areas are covered by glaciers. (c) shows the difference between LGM$_{\mathrm{ALPSLESS}}$ and LGM topography. The boxes in (a) and (b) are the two regions used for the Skew–T diagrams: Site A represents the north-western and site B the central-southern region.

**Table 2.** Set of experiments carried out in this study. The first column indicates the name of the WRF simulation, the second column the perpetual conditions, the third column the northern hemispheric ice sheets (this includes the modifications in the driving global model), the fourth column the FIS, the fifth column the Alpine glaciers, the sixth column the land cover, and the seventh column the length of the simulation.

| Name | Perpetual conditions | North Hemis. ice sheets | Fennoscandian ice sheet | Alpine glaciers | Land cover | Simulation length |
|---|---|---|---|---|---|---|
| PD$_{PD}$ | 1990 | 1990 | 1990 | 1990 | 1990 | 21 years |
| LGM$_{LGM}$ | LGM | LGM | LGM | LGM | LGM | 21 years |
| MIS4$_{LGM66}$ | MIS4 | 66 % LGM | 66 % LGM | LGM | LGM | 21 years |
| MIS4$_{LGM}$ | MIS4 | 100 % LGM | 100 % LGM | LGM | LGM | 21 years |
| MIS4$_{LGM125}$ | MIS4 | 125 % LGM | 125 % LGM | LGM | LGM | 21 years |
| LGM$_{FIS50}$ | LGM | LGM | 50 % LGM | LGM | LGM | 12 years |
| LGM$_{FIS150}$ | LGM | LGM | 150 % LGM | LGM | LGM | 12 years |
| LGM$_{ALPSLESS}$ | LGM | LGM | LGM | reduced LGM | LGM | 21 years |

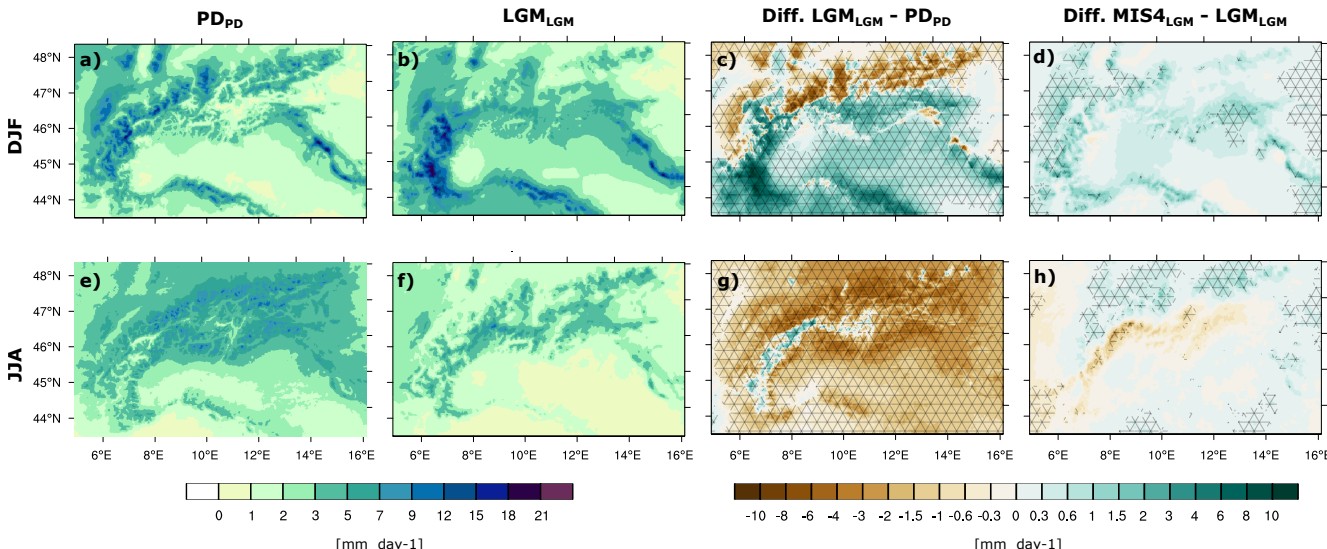

**Figure 3.** Climatological mean values of daily precipitation intensity over the Alps for (a, b, c and d) winter (DJF) and (e, f, g, and h) summer (JJA): (a, e) the mean PD precipitation, (b, f) the mean LGM precipitation, (c, g) the difference between LGM and PD and (d, h) the difference between MIS4 and LGM. Crosshatched areas represent statistically significant differences with a significance level of 0.05 (using a two-tailed bootstrapping technique).

## Site A : North-Western Region

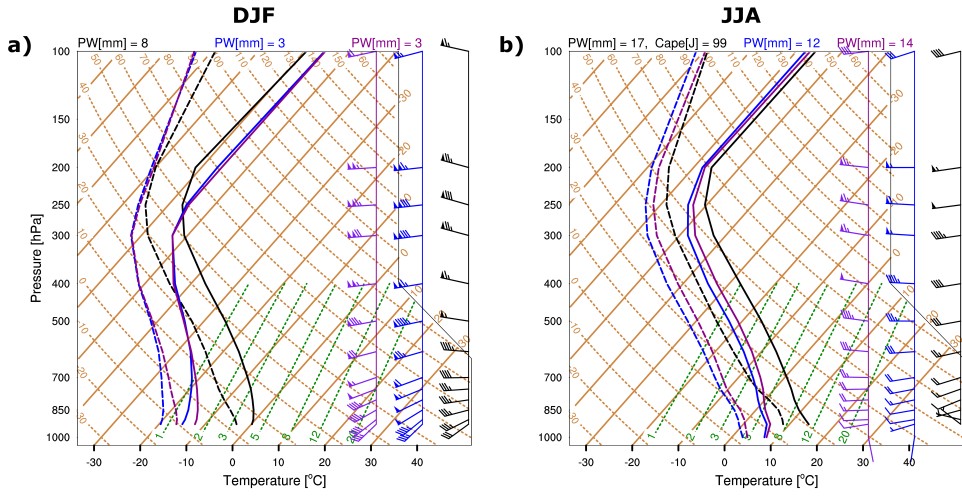

## Site B : Central-Southern Region

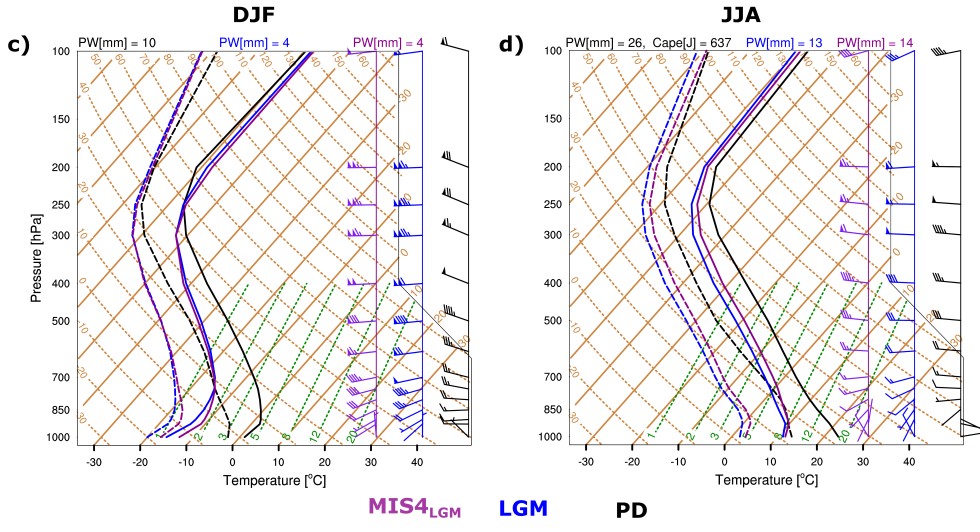

**Figure 4.** Skew–T diagram for site A (north-western region; a and b) and site B (central-southern region; c and d). Sites' locations are shown in Fig. 2. (a and c) represent climatological vertical profiles for DJF and (b and d) for JJA. $MIS4_{LGM}$ climate is illustrated by purple lines, LGM climate by blue and PD climate by black. There are two vertical profiles for each climate: air temperatures in solid lines and dew-point temperatures in dashed lines. Skewed straight dashed green lines represent isohumes labelled on the bottom of the diagram. The mixing ratio is obtained from crossing the saturated mixing ratio lines crossed with the dew-point temperature vertical profile and its value increases to the right at a constant pressure level. Solid brown lines are isotherms. Dry adiabatic lines (lines of equal potential temperature $\theta$) are slightly curved dashed brown lines. In the wind bars, the triangle, line and half-size line represent 50, 10, 5 km $h^{-1}$, respectively. Furthermore vertically integrated precipitable water (PW) and CAPE are given at the top of each panel. Note that CAPE is only displayed if it is different from zero. Please see Sect. 3 for a in-detail description of the Skew–T diagram.

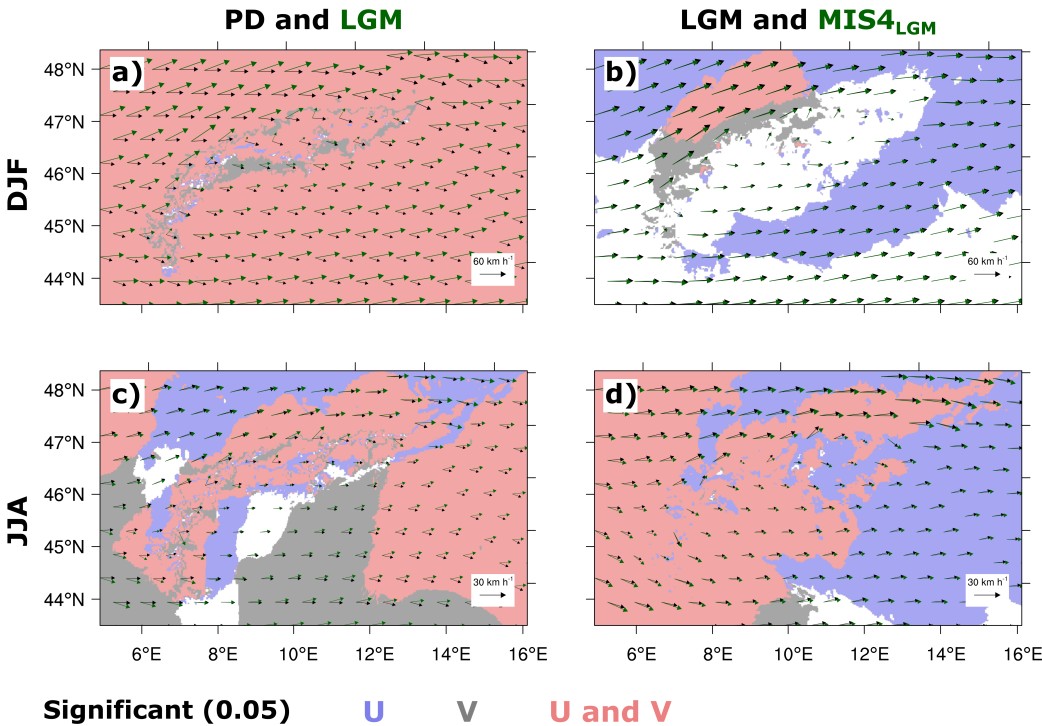

**Figure 5.** Climatological mean wind vectors over the Alps for (a and b) DJF and (c and d) JJA: (a and c) black and green wind vectors correspond to PD and LGM, respectively, (b and d) black and green wind vectors correspond to LGM and MIS4, respectively. Red shading illustrates statistically significant differences in zonal (U) and meridional (V) wind components with a significance level of 0.05 (two-tailed bootstrapping technique), blue and grey shading indicate significance either in the U or V wind component, respectively. Please note that the reference wind vectors differ for DJF and JJA.

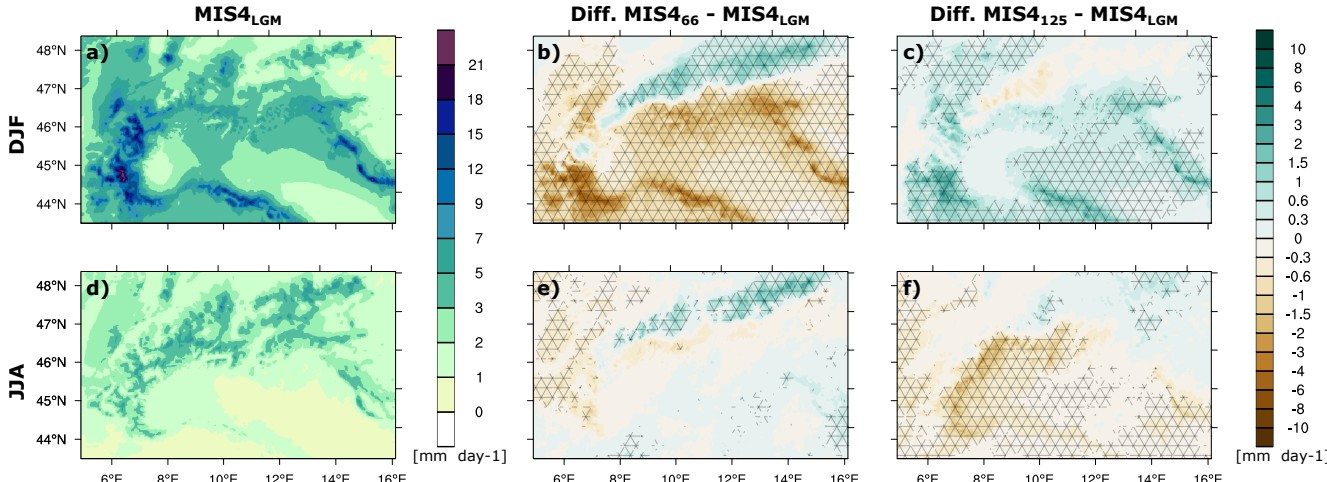

**Figure 6.** Climatological mean values of daily precipitation intensity over the Alps. (a) represents $MIS4_{LGM}$ precipitation for DJF, (b) the differences between $MIS4_{LGM66}$ and $MIS4_{LGM}$, (c) as (b) but between $MIS4_{LGM125}$ and $MIS4_{LGM}$. (d), (e) and (f) as (a), (b) and (c) but for JJA. Crosshatched areas indicate statistically significant differences at a significance level of 0.05 (using a two-tailed bootstrapping technique).

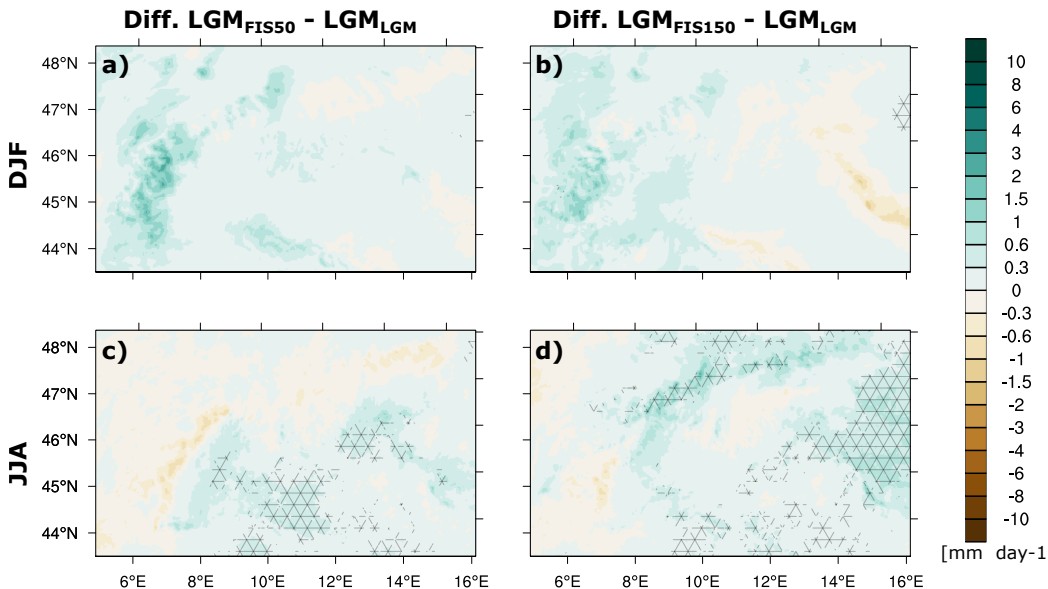

**Figure 7.** Differences in climatological mean values of daily precipitation intensity over the Alps for (a and b) winter (DJF) and (c and d) summer (JJA): (a and c) LGM$_{FIS50}$ minus LGM$_{LGM}$ and (b and d) LGM$_{FIS150}$ minus LGM$_{LGM}$. Crosshatched areas indicate statistically significant differences at a significance level of 0.05 (using a two-tailed bootstrapping technique).

# Site A : North-Western Region

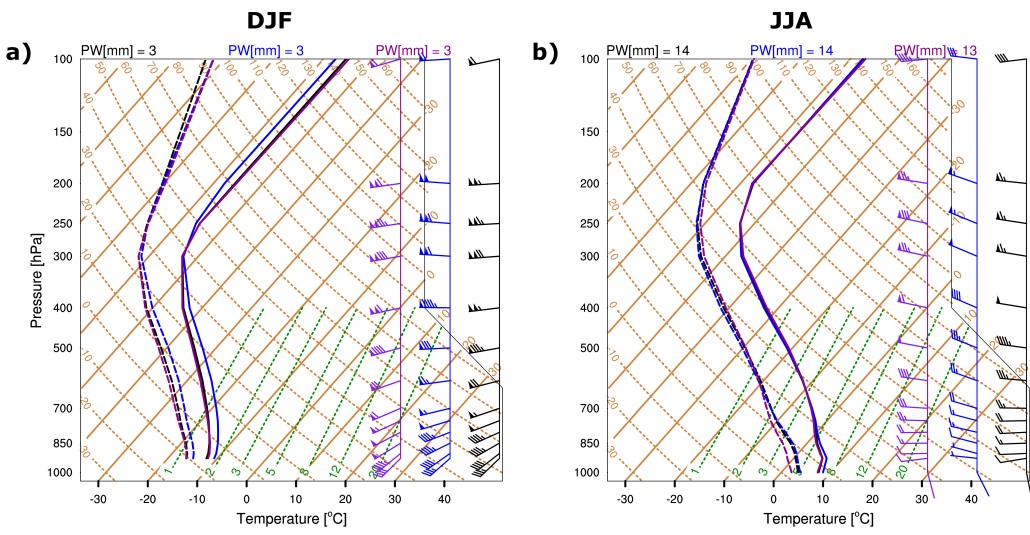

# Site B : Central-Southern Region

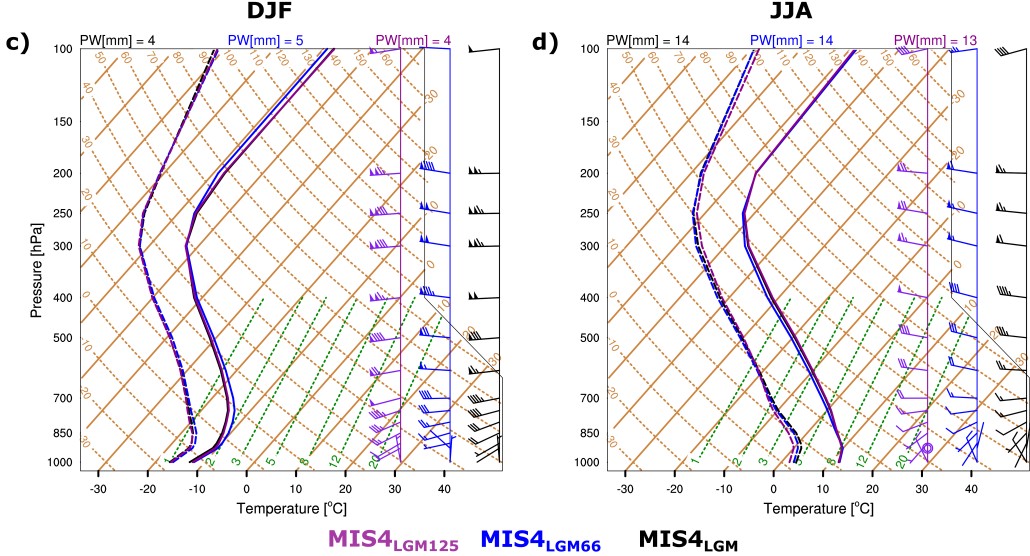

**MIS4**$_{\text{LGM125}}$   **MIS4**$_{\text{LGM66}}$   **MIS4**$_{\text{LGM}}$

**Figure 8.** As Fig. 4, but for MIS4$_{\text{LGM125}}$ (purple), MIS4$_{\text{LGM66}}$ (blue) and MIS4$_{\text{LGM100}}$ (black)

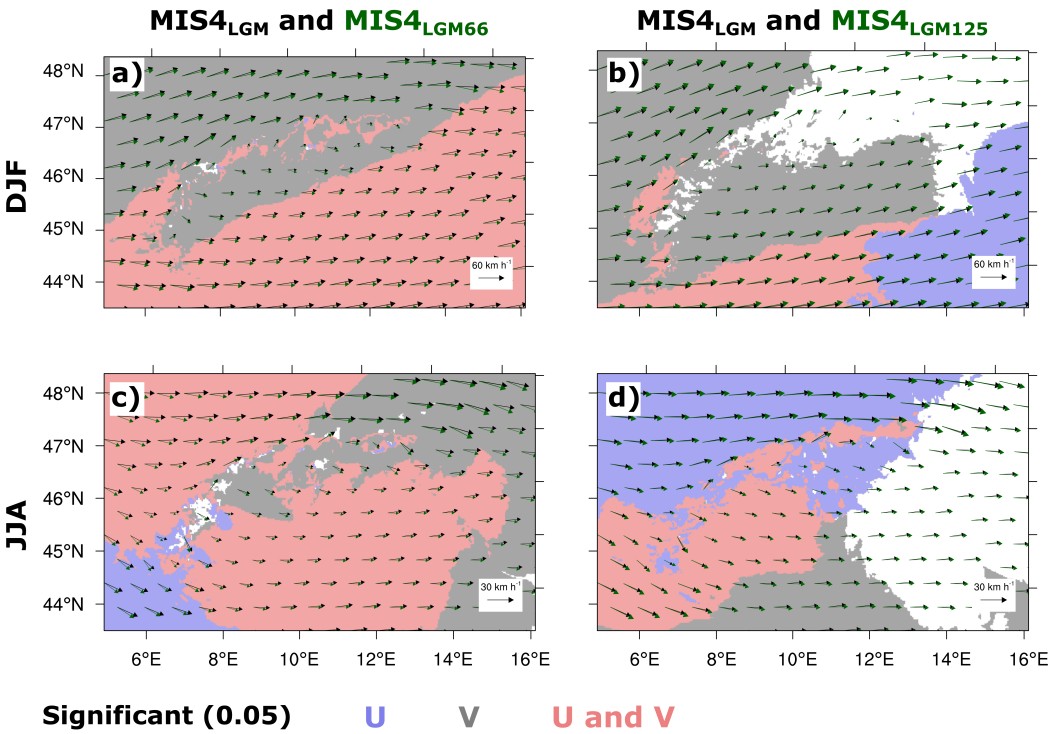

**Figure 9.** Climatological mean wind vectors over the Alps. (a) represents wind vectors for DJF, black and green vectors correspond to MIS4$_{LGM}$ and MIS4$_{LGM66}$, respectively, (b) as (a) but green vectors to MIS4$_{LGM125}$. (c) and (d) as (a) and (b) but for JJA. Red shading indicates statistically significant differences in zonal (U) and meridional (V) wind components at a significance level of 0.05 (two-tailed bootstrapping technique), blue and grey shading as the red one but only in U and V wind components, respectively. Please note that the reference wind vectors differ for DJF and JJA.

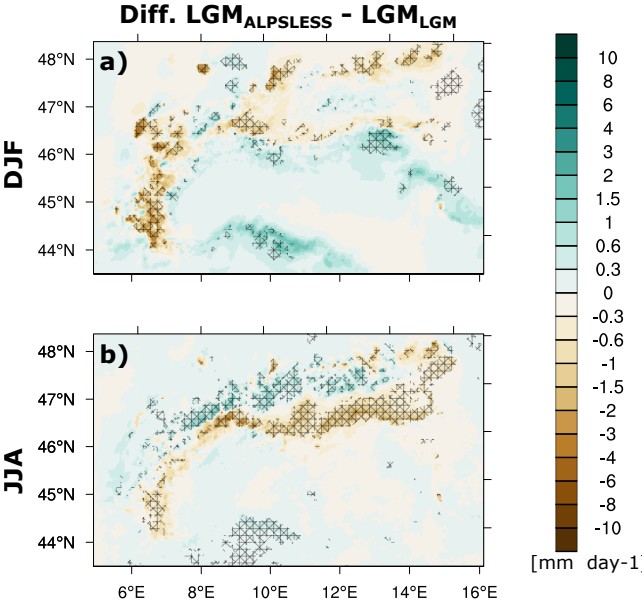

**Figure 10.** Differences in climatological mean values of daily precipitation intensity over the Alps between LGM$_{\text{ALPSLESS}}$ and LGM$_{\text{LGM}}$ for (a) winter (DJF) and (b) summer (JJA). Crosshatched areas indicate statistically significant differences at a significance level of 0.05 (using a two-tailed bootstrapping technique).

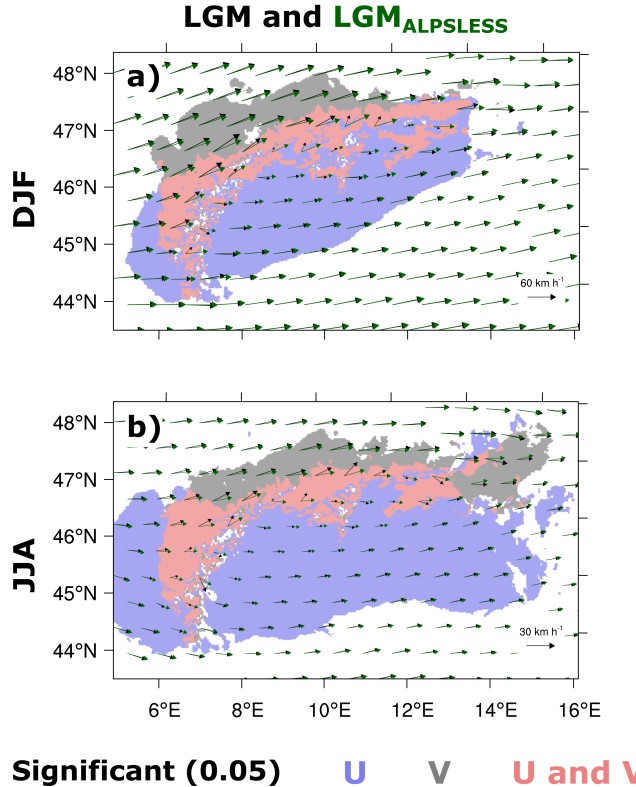

**Figure 11.** Climatological mean wind vectors over the Alps. (a) represents wind vectors for DJF, black and green vectors correspond to LGM$_{\text{LGM}}$ and LGM$_{\text{ALPSLESS}}$, (b) as (a) but for JJA. Red shading indicates statistically significant differences in zonal (U) and meridional (V) wind components at a significance level of 0.05 (two-tailed bootstrapping technique), blue and grey shading as the red one but only in U and V wind components, respectively. Please note that the reference wind vectors differ for DJF and JJA.