# Peer review of "The role of ice-sheet topography in the Alpine hydro-climate at glacial times"

_Climate of the Past, 2021_

## Author Comment (AC1)

**Final Response to Referee #1**

We thank the reviewer Maria Fernanda Sanchez Goñi for the careful and thorough reading of our manuscript. The comments have been carefully considered and responded. Please find below our response to each comment.

**General comments**

1. The manuscript submitted by Velasquez et al. to the Climate of the Past presents a combination of global and regional model simulations to test the sensitivity of the glacial Alpine hydro-climate to northern hemisphere, Laurentidae and Fennoscandian, and local ice-sheet changes during the Last Glacial Maximum (LGM) and Marine Isotope Stage (MIS) 4. For the LGM, they find that thickening of the northern hemisphere ice-sheets, mainly the Laurentidae ice caps, and local ice-sheet topography generally lead to increase in winter precipitation and decrease in summer rainfall, both enhancing glacial conditions. In winter, dynamics processes related to the intensity and position of the Alpine winds explain the moistening in the southern part of the Alps while the simulated summer drying all over the Alps is related to thermodynamic processes, i.e. colder temperatures. In contrast, Fennoscandian ice-sheet changes have a negligible impact on the Alpine hydro-climate. For MIS 4, marked by lower global ice volume than the LGM, Velasquez et al. find wetter climate in the Alps attributed to thermodynamic processes, i.e. warmer temperatures. This manuscript is clearly written and convincing for the LGM. In contrast, I have several caveats related to the MIS 4 model results and comparison with the regional (western European) climate at that time when compared to that of the LGM (see below). Overall, this work deserves publication in CP after the authors address the comments that I have listed below. I am not a modelling expert, and I will only comment on the data discussed in this work.

**RESPONSE:**

We thank you for your detailed comments. We will take care of the concerns related to analysis of the MIS4 climate and the comparison with the LGM in the following responses and in the revised manuscript.

**Major comments**

2. Lines 40-50: It would be relevant to cite the paper by Harrison and Digerfeldt (1993, Quaternary Science Reviews), one of the first paper showing that southern Europe was wet during the LGM (centered at 21 ka) based on the high water levels recorded in several lakes around the Mediterranean region. Iberian margin pollen records also provide evidence that the LGM in southern Europe was wetter than the Heinrich Stadial (HS) 2 and HS 1 bracketing it (Naughton et al., 2007, Marine Micropaleontology; Turon et al., 2003, Quaternary Research).

**RESPONSE:**

We agree that some pollen-based reconstructions show a wetter southern Europe compared to present day. Therefore, we will reformulate these lines (lines 40-50) and add this information in the revised version of the manuscript. Additionally, we will mention other studies that highlight the wetter conditions in southern Europe compared to other periods, e.g. Heinrich Stadials and Marine Isotope Stages. The changes will be done as follows:

"... The same data is also used to reconstruct the hydro-climatic response over Europe at LGM mainly showing drier conditions over Europe (reduction in precipitation of around 200 mm year-1) compared to PD (Wu et al., 2007; Bartlein et al., 2011). Roucoux et al. (2005) indicated that the LGM was not necessarily a dry period everywhere in Europe. For instance, Turon et al. (2003); Roucoux et al. (2005); Naughton et al. (2007) and Ludwig et al. (2018) suggested that southern Europe was wetter compared to the rest of Europe and also to adjacent periods, i.e. the Marine Isotope Stage 3 (Voelker et al., 1997; Kreveld et al., 2000) and to the Heinrich event 1 and 2 (Sanchez Goñi and Harrison, 2010; Álvarez-Solas et al., 2011; Stanford et al., 2011). Compared to PD, many studies suggest that the wetter conditions in southern Europe can be explained by a southward shift in the North Atlantic storm track during the LGM (e.g. Hofer et al., 2012a; Luetscher et al., 2015; Merz et al., 2015; Ludwig et al., 2016; Wang et al., 2018; Raible et al., 2020; Lofverstrom, 2020). This southward shift is in line with other climate reconstructions that suggest circulation-induced changes in the moisture transport. For instance, Harrison and Digerfeldt (1993) found very different patterns of lake-level changes across Europe suggesting major changes in European atmospheric circulation patterns. Other climate reconstructions also suggest circulation-induced changes in the moisture transport (Florineth and Schlüchter, 2000). In this case, the atmospheric circulation is..."

3. Lines 50-54 and lines 407-415: To support the idea that MIS 4 was warmer and wetter than the LGM, Velasquez et al. only refer to global studies (Eggleston et al., 2016), Australasian records (De Deckker et al., 2019; Newham et al., 2017) and model simulations for the North Atlantic and Greenland climate (Hofer et al., 2012; Merz et al. papers) that cannot be used to account for the climate in Europe at that time and, particularly, at 65 ka, the date chosen for their simulations. This date is concomitant with the maximum of global ice volume during MIS 4 (Waelbroeck et al., 2002, Quaternary Science Reviews), coincides, within the chronological uncertainties, with Greenland Interstadial 18 and precedes the massive iceberg discharges in the North Atlantic leading to the HS 6, 64-60 ka (Sanchez Goñi et al., 2013, Nature Geoscience, Figure S3 of the supplementary information).

To realistically compared the recorded and simulated climate in Europe at 65 ka, the authors should discussed their wind field and climate reconstructions in the context of the climate prevailing in the western European margin and the adjacent landmasses during this period, climate that is mainly controled by the westerlies during winter. The work by Sanchez Goñi et al. (2013, Nature Geoscience, Figure 2 and Figure S3 of the supplementary information) zooms in on MIS 4, and shows relatively wet and warm atmospheric conditions at 65 ka, based on the increase of heathlands and pine forest, contemporaneous with foraminifera-based warm summer sea surface temperatures in the western European margin, reaching 15°C in the Bay of Biscay and the SW Iberian margin and 10°C in the NW Iberian margin. However and in contrast with the authors' idea that the LGM was colder than MIS 4 in the European margin, higher sea surface temperatures in the Bay of Biscay (Sanchez Goñi, 2020, Evolutionary Human Sciences, Figure 2) and in NW and SW Iberia (Sanchez Goñi et al., 2008, Quaternary Science Reviews, Figures 3 and 4) are recorded during the LGM compared to MIS 4. Both periods are characterised by low and similar temperate forest abundance and similar heathlands development suggesting that MIS 4 was not warmer and wetter compared to the LGM.

**RESPONSE:**

We agree that the description of MIS4 is too short and the citations do not focus on the Europe climate. We thank the reviewer for highlighting these studies. In the revised manuscript, we will further introduce the MIS4 climate and briefly discuss its uncertainties. Therefore, we will include the studies of Sánchez Goñi et al. (2008) and Sánchez Goñi (2020) and we will also reformulate these lines and add more information as follows:

Lines 50-54:

"...MIS4 climate is less understood compared to LGM as proxy data availability is further reduced compared to the LGM. Globally, available paleoclimate reconstructions characterise MIS4 to be warmer than the LGM (e.g. Eggleston et al., 2016; Newnham et al., 2017; De Deckker et al., 2019) with a sea level drop of roughly 80 m compared to PD (e.g. Cutler et al., 2003; Siddall et al., 2008, 2010; De Deckker et al., 2019). Focusing on Europe, MIS4 shows relatively wet and warmer conditions at 65 ka compared to the period 85-50 ka (Sánchez Goñi et al., 2013). Still, Sánchez Goñi et al. (2008) and Sánchez Goñi (2020) found drier and colder conditions around the Iberian Peninsula during MIS4 compared to the LGM. This might suggest, similar to the LGM period, that the MIS4 climate was not necessarily homogeneously wetter and warmer across Europe."

Lines 407-415:

"...The MIS4 (MIS4LGM) climate shows enhanced winter precipitation compared to the LGM. The reason is that the MIS4 climate state is in general warmer (Hofer et al., 2012a,b; Merz et al., 2013, 2014a,b, 2015, 2016) and thus more moisture is globally available. Wind changes do not contribute to these wetter conditions in the Alpine region as they become weaker and therefore reduce the orographically forced uplifts, which also suggests an overall reduction of the moisture transport to the Alps. Thus, we interpret the winter changes between MIS4 and LGM to be purely thermodynamically driven (Clausius–Clapeyron equation). In summer, MIS4LGM shows slightly wetter conditions at the northern face and drier conditions at the southern side of the Alps. The northern wetter conditions are induced by an increase in the tropospheric vertical wind shear enhancing convection processes. The drier conditions at the southern face of the Alps are explained by slightly clockwise rotated winds, which enhance the Foehn effect. Thus, dynamical processes are also relevant to explain the summer precipitation changes between MIS4 and LGM."

4. Line 395-406: The authors should add in the revised version of the manuscript the new evidence from a cryogenic carbonate record in the Alps (Spötl et al., 2021, Nature Comm.) showing heavy snowfall during autumn and early winter during the LGM. These results combined with thermal modelling, provide compelling evidence that the LGM glacier advance in the Alps was fuelled by intensive snowfall late in the year, likely sourced from the Mediterranean Sea.

**RESPONSE:**

We appreciate the reviewer to bring to our attention this study. We will include this new evidence in the discussion and conclusions part of the revised version.

5. Lines 436-440: The authors should delete the references of Finlayson et al., 2004, 2006 and 2008 and that of Burke et al., 2014 and Baena Preysler et al., 2019. These works do not refer to the Alpine

**regions and, therefore, they are not relevant for this study.**

**RESPONSE:**

We will delete these references in the revised version of the manuscript.

**Minor comments**

m1. Line 72: Please add Regional Climate Models to explain RCM.

**RESPONSE:**

We will add it in the revised version of the manuscript.

**m2. Line 158: Please replace « «eighth experiment » with « eighth experiments ».**

**RESPONSE:**

We cannot understand your suggestion as we refer to a single experiment in this sentence, i.e. the experiment number eight. To clarify any misunderstanding, we will reformulate this line in the revised manuscript as follows:

"...In the experiment number eight, we investigate..."

**m3. Line 255: Please replace « associated to » with « associated with ».**

**RESPONSE:**

We will replace it in the revised version of the manuscript.

We would like to thank the reviewer Maria Fernanda Sanchez Goñi for the time invested in reviewing the manuscript so carefully. We are looking forward to meeting her/his expectations.

Best regards,

Patricio Velasquez (on behalf of the author team)

**References**

[revised manuscript text omitted]

---

## Author Comment (AC2)

**Final Response to Referee #2**

We appreciate the time the reviewer has invested in reading the manuscript in such a careful and thorough manner. The comments have been carefully considered and responded. Please find below our response to each comment.

**General comments**

**1.** *An interesting study investigating the sensitivity of the glacial Alpine hydro-climate to northern hemispheric and local ice-sheet changes, using the chain of GCM-RCM simulations to perform comparison and sensitivity analysis for two glacial periods, the LGM and MIS4, with the different ice-sheet thickness (in different glacial regions) effects on the Alpine hydro-climate conditions. The results are analyzed in very much detail, although in some cases the effects are rather small and with respect to the simulation concept, I would not be so sure that the conclusions are so firm (see some further more detailed comments).*

**RESPONSE:**

We thank you for your detailed comments. We will take care of the concerns related to the conclusions in the following responses and in the revised manuscript.

**Major comments**

**2.** *First, to make these more strong at least some small ensemble (a few models) should be employed to see the robustness of the results.*

**RESPONSE:**

We agree that an ensemble can increase the robustness of the results, especially when analysing the climate simulations separately. However, we think that this suggestion is beyond the scope of our manuscript as this study focuses on topography-related sensitive experiments rather than assessing the uncertainties of the climate simulations. Moreover, we would like to mention that each regional climate simulation implies very high computational costs, e.g. calculation time and storage and our current systems at use do not allow for further simulations. Note further that we would also need to perform global model simulations. If the reviewer suggests to use different model chains, i.e., another GCM and another RCM, then this is certainly beyond the capacity of the group as this would imply gaining knowledge and experience with these models. Even modelling centres only use and maintain one model. Our guess is that the reviewer has a community effort in mind, so that a small ensemble would rather orientate within the modelling community, e.g. CORDEX (CORDEX, 2019). Clearly, we agree that this would be beneficial, but to our knowledge regional modelling in the paleo perspective is rather new. We know only two other groups working on paleo issues

and none has used convection permitting scales (except our group). Therefore, we see this study as a starting point and the robustness is given by statistical tests. We will try to be more specific that the results obtained and discussed may depend on the model chain used.

**3.** *Second, the chain of the model domains is a bit strange. In my opinion, the innermost domain is too small to develop properly the circulation in the vicinity of the Alpine region and due to its location at the edge of further domain boundaries from all the three sides, the discussion of the results of simulation on the north-west and south-east sites are not equally valid. What comes from the south is more or less based on the 18 km resolution domain as with respect to the proximity to the other domains edges there is no enough room and time to properly develop in CP resolution. This might be of importance as there is a significant change of land-use in Adriatic till this border, as well as with respect to the shift of polar front under glacial conditions. I understand the limitations of these extensive and demanding simulations, however, these aspects should appear some way in the presentation and discussion of the results, with the limitations clearly declared and possible uncertainties pointed out.*

**RESPONSE:**

We appreciate that the reviewer brought to our attention that the model setup is still a bit unclear. We would like to mention that there is a relaxation zone that refers to the lateral areas of the domain where the WRF model is nudged or relaxed towards the larger-scale input data, i.e. the lateral boundary conditions. We used a relaxation zone of 5 grid points (10 km in the innermost domain D4) at each edge which is deleted before the analysis, i.e. the figures in manuscript do not show the relaxation zone. Without this relaxation zone in the innermost domain, there are still around 60, 80, 120 and 80 km distance (30, 40, 60 40 grid points, respectively) from the first elevated areas (e.g. above 1500 m a.s.l.) to the west, north, east and south edges of the domain, respectively. Furthermore, the domain D3 in 6 km resolution also uses a 5 grid point relaxation zone and is convection permitting. Checking the simulations we find convection-type structures emerging also from the south, which where also found in simulations with the same setup but driven by reanalysis data (Gómez-Navarro et al., 2015, 2018). Still, we agree that a wider region would be beneficial. In the revised manuscript, we will clarify this in the method section and we will also mention it in the conclusion part.

**4.** *Further, concerning the discussion of the relative humidity, I would like to stress the dependence of it on the actual temperature itself as well, which can be quite significant when comparing the PD and LGM, see below in specific comments, but please check throughout the paper. The same distance between the dew point and actual temperatures under these different conditions of PD and LGM will not mean the same relative humidity.*

**RESPONSE:**

We fully agree that the relative humidity also depends on the air temperature which is strongly different between PD and LGM. Also, we think that the distance between the dew point and air temperature would poorly represent the relative humidity on a linear-scale diagram, not even a qualitatively representation. However, we believe that the logarithmic scale on the SkewT diagram allows to interpret the distance between the dew-point temperature and air temperature as a qualitative representation of the relative humidity. We will take care of this concern in the following responses and in the revised version the manuscript.

**5.** *Moreover, concerning the mixing ratios in the diagram, that means saturated mixing ratio under the actual atmospheric conditions, which is not saying too much about the precipitable water, it depends on the relative humidity. Connection to Clausius-Clapeyron equation makes more sense in the discussion of extreme precipitation, where really different temperature of the atmosphere with different maxima of potential mixing ratios results in different amount of precipitable water. However, in the results presented the relative humidity looks to be rather lower.*

**RESPONSE:**

We will change the name of the mixing ratio lines (page 6 lines 178) to saturated mixing ratio lines in the revised manuscript. Additionally, we will modify the following lines (page 6 lines 185-186):

"...dew-point temperatures (dashed lines). The latter simply indicate temperatures at which the air becomes saturated and is used to deduce the mixing ratio with height, i.e., the amount of water vapour in the air where the dew point temperature line crosses the mixing ratio line."

to

"...dew-point temperatures (dashed lines). The temperatures are used to deduce the mixing ratios with height, i.e., the amount of water vapour in the air with height, whose values are obtained from the saturated mixing ratio lines when they are crossed by the temperature vertical profiles. The dew-point temperatures indicates the temperatures at which the air becomes saturated."

Regarding the precipitable water (PW), also called integrated water vapour (IWV), we agree that the PW is not directly represented by the mixing ratio at one pressure level (value of mixing ratio is obtained from the saturated mixing ratio line when it is crossed by the air temperature vertical profile). The PW represents the amount of water vapour of an atmospheric column expressed as the depth of water if that vapour were condensed, i.e. water vapour available for precipitation. The PW value is calculated by vertically integrating the water vapour mixing ratios across height and it does not depend on the temperature (Eq. 1; e.g. Rozsa, 2012; González-Rojí et al., 2018). These values are shown at the top of each panel of figure 4 and 8 in the manuscript.

$$PW \ = \ IWV \ = \ \frac{1}{g} \int_{sfc}^{top} q \, dp, \tag{1}$$

where $g$ is the gravitational acceleration, $q$ the vapour mixing ratios, $top$ and $sfc$ the pressure at the uppermost and lowermost model level, respectively. We will clarify this in the method's and result's section of the revised version of manuscript.

**6.** *A formal comment concerns the rotation of the wind (appearing throughout the paper). I would recommend using the term turning of the wind, which is commonly used when describing the changes of wind direction with height, i.e. wind turns clockwise or anticlockwise. Similarly, something can cause to turn the wind to some direction, e.g. changes of ice-sheet heights, LGM conditions etc.*

**RESPONSE:**

We appreciate this suggestion. We have changed the term "rotation" to "turning" in the revised manuscript.

**Specific comments**

**A1.** *Page 1, line 7: explained*

**RESPONSE:**

It will be changed.

**A2.** *Page 4, line 126-9: Actually, this simulation strategy is not too rigorous comparing what is commonly required from RCM simulations*

**RESPONSE:**

We partly agree with the reviewer. A better simulation strategy would generally be when a RCM runs for 30 years in a single piece with a spin-up period longer than 2 months. However, we think that this better strategy would hardly influences our conclusions, especially when our analyses are completely based on mean values, i.e. the climatology. We would like to highlight the reason of choosing our strategy in the following.

Splitting up the simulations is explained by the time-consuming setup to run a simulation over the Alps at 2 km resolution. Namely, 3 model years are equivalent to 1 month in real time, which means that a 21-years simulation in a single piece would have taken at least 7 months in real time without any interruption. Each regional climate simulation is forced under perpetual conditions, i.e. constant climate conditions, which allows to split it into 3-years simulations that represent the same climate state. For example, a 21-years simulation is performed under perpetual 1990 conditions. To that end, the 21-years simulation is split into seven individual 3-years simulations which all represent the same 1990 climate conditions. We will briefly mention the common strategy and reformulate these lines in the revised version of the manuscript as follows:

"...simulation segments, respectively. Note that the reason of splitting is to efficiently use the available computer facilities (similar to accompanying studies such as Velasquez et al., 2020, 2021), even though regional climate simulations would commonly be performed in one single piece. For each segment, a 2-month spin-up ..." .

**A3.** *Page 6, line 171: not clear how 30 annual mean samples can be selected from 21 or even 12 years simulations?*

**RESPONSE:**

We thank the reviewer for this comment. The number "30" is a mistake. It should not have been written and it will be deleted in the revised version of the manuscript.

**A4.** *Page 6, line 188-9 the relative humidity still depends on the actual temperature as well, the same difference between the actual and dew point temperature does not imply the same relative humidity, especially under quite different temperatures like for PD and LGM*

**RESPONSE:**

We fully agree that relative humidity (RH) depends on the actual air temperature as well. This dependency is implicitly included in the SkewT diagram and therefore the values of RH can be qualitatively evaluated by the distance between the temperatures profiles, i.e. the distance between the dew-point temperature and air temperature. We will clarify this in the revised version of the manuscript as follows:

Lines 186-189:

"...Both temperatures are used to investigate the relative humidity, i.e., the level of saturation at a certain pressure and temperature. Note the SkewT diagram implicitly includes the dependency of the relative humidity on temperature by the logarithmic scale of the saturated mixing ratio lines. Therefore, the relative humidity can be obtained by qualitatively estimating the distance between both temperatures profiles. A short distance indicates a high relative humidity and, inversely, a large distance a low relative humidity."

**A5.** *Page 8, line 230: In fact, mixing ratios are not shown, in the diagram, there are saturated mixing ratios for the given conditions provided, precipitable water depends on actual relative humidity*

**RESPONSE:**

We will change the name of these lines in the revised manuscript. Also, we have clarified the concern about the relative humidity in a previous answer. Please refer to response to the fifth major.

**A6.** *Page 8, line 231-2: relative humidity is not the same with the same difference between the dew point temperature and actual temperature, it depends on temperature itself as well (see above)*

**RESPONSE:**

We have clarified this concern in a previous answer. Please refer to response to the fifth major.

**A7.** *Page 8, line 244: I do see boundary layer in PD as well, of course, clearly, with less stable lapse rate due to surface warming in summer*

**RESPONSE:**

We agree that there is a visible boundary layer in PD as well with less stable lapse rate due to surface warming in summer. We will modify this line as follows:

"...layer under PD and LGM conditions, particularly clear during the LGM."

**A8.** *Page 8, line 249: Actually, for the wind field (circulation) analysis rather larger scale (domain) should be shown with pressure fields changes, which will be probably a stronger driver of the circulation. The alpine effect can be seen just in the close proximity, where even in 700 hPa level Alps can create barrier with different height during PD and LGM (MIS4) inducing either overflow or flow around some parts, which can be well resolved in the highest resolution (in connection to stability as well)*

**RESPONSE:**

We would like to mention that the PD and LGM CCSM4 simulations have been analysed over Europe in a variety of studies, including additional simulations for other glacial and interglacial states (e.g. Hofer et al., 2012a,b; Merz et al., 2013, 2014b,a, 2015, 2016; Landais et al., 2016). These study particularly focused on changes in the atmospheric circulation during glacial times. Compared to PD conditions, the LGM simulation reveals a clear southward shift, a more zonal orientation of the storm track over the North Atlantic and substantial changes in the weather patterns (Hofer et al., 2012b,a). These changes are able to explain precipitation anomalies over Europe, especially over the Iberian Peninsula and the western part of the Mediterranean Sea.

Nevertheless, we will include a brief analysis of the larger-scale sea level pressure and wind fields at 700 hPa in the revised version of the manuscript to further examine the drivers of the changes in the Alpine Hydro-climate.

**A9.** *Page 9, line 279: Actually, I do not see so much significant difference (despite formal statistical significance) to discuss here except the cases of the proximity of Alpine ridge, where the differences can be due to different heights of the terrain (glaciers), as mentioned above*

**RESPONSE:**

The reviewer refers to Fig 5. There, the Alpine ice sheet is not changed so the changes are not due to topographic differences. We agree that the wind vectors are not suggesting a similar strong changes in Fig 5d than e.g. Fig 5a. Still, the shading shows the statistical significance of the changes. Maybe a problem is the colours of the wind vectors so we will try to redesign the figure.

**A10.** *Page 10. Line 319: it is rather tiny*

**RESPONSE:**

As answered in previous response, we will face this concern in the analysis of the larger-scale of wind fields at 700 hPa in the revised version of the manuscript.

**A11.** *Page 10, line 321: This will be really negligible, especially with respect to the relative humidity, which will be rather low. By the way, again, the green dashed lines represent saturated mixing ratios, not actual mixing ratios.*

**RESPONSE:**

As in response to fifth major comment, we would like to mention that a mixing ratio is obtained from the saturated mixing ratio line when it is crossed by the temperature vertical profile. Regarding the definition

of precipitable water, please refer to the same response (fifth major comment). We will reformulate this line in the revised version of the manuscript as follows:

"...atmosphere. The latter results in a small increase of moisture availability in the middle-to-low atmosphere (dashed green lines crossed by the air temperature profile), especially in the central-southern region (site B; Fig. 8c) where there is more precipitable water (PW values at the top of Fig. 8a and c).

**A12.** *Page 10, line 325: As above, saturated mixing ratios*

**RESPONSE:**

The name will be changed in the revised manuscript.

**A13.** *Page 10, line 328: glacial climate conditions in Alpine region*

**RESPONSE:**

We thank the reviewer. This will be changed in the revised version of the manuscript.

**A14.** *Page 11, line 335-6: Again, the changes are really very tiny, despite statistical significance*

**RESPONSE:**

We agree that the differences are visually tiny. As mentioned in the response to the minor comment A9, we will face this concern in the analysis of the larger-scale of wind fields at 700 hPa in the revised version of the manuscript.

**A15.** *Page 11, line 354: Actually, it is difficult to see the significance in such a small region*

**RESPONSE:**

In the revised version of the manuscript, we will change the visualisation of the significance to another drawing pattern.

**A16.** *Page 11, line 355-7: Actually, direct westerly inflow is more or less perpendicular to the barrier of the Alpine ridge in the western part, while in the eastern part it is rather parallel.*

**RESPONSE:**

We agree with the reviewer. In these lines, we referred the precipitation changes to the modification of ice-sheet thickness rather than the wind changes. To clarify this, we will add more information to these lines in the revised version of the manuscript as follows:

"...Interestingly, the reduction in precipitation is higher in the western part than in the central to eastern part of the Alps, although the Alpine ice-sheet thickness is reduced more strongly in the east than in the west (Fig. 2c). This cannot directly be explained by the changes in the wind field around the Alps at 700 hPa, which are caused by the reduction of the Alpine ice-sheet thickness. The reason is that we would expected wetter conditions in the western part since the winds slightly turn clockwise (more perpendicular to the Alps) and slightly stronger."

**A17.** *Page 12, line 371-2: Actually, despite the formal statistical significance I do not see so big changes except in the close proximity of the mountain ridge (as noticed above), where the direct interaction of the changed top of the barrier is evident and causing direct changes in flow patterns on that level.*

**RESPONSE:**

We agree with the reviewer. We will reformulate these lines in the revised manuscript as follows:

"...The LGM$_{ALPSLESS}$ winds are a bit stronger and slightly turn clockwise compared to LGM$_{LGM}$. The slight turning is associated with significant changes in V at the northern face of the Alps, in both wind components over the Alpine axis and in U in the south of the Alps (Fig. 11a)."

**A18.** *Page 12, line 374-7: However, a strong issue is how the model represents the transfer of precipitation from the place of creation downstream with the flow to the place it is considered as reaching the surface.*

**RESPONSE:**

Our guess is that the reviewer suggests to perform a back trajectory analysis to identify where the precipitation is coming from, e.g, North Atlantic, land surface over Europe (water recycling) and the Mediterranean. Clearly, this would be a interesting question, but we think that this is beyond the scope of this study but certainly a next step to be mentioned at the end of this manuscript.

**A19.** *Page 13, line 405-6: This would be nice shown on the analysis of convective precipitation*

**RESPONSE:**

We would like to mention that this analysis is not possible as our simulations are performed using convection-permitting in the two innermost domains (6 and 2 km). This does not allow to separate the convective precipitation from the total precipitation since it is directly resolved in the model. CAPE is another quantitative measure that refers to convection activity. CAPE values are shown at the top of each panel on figure 4 and 8, when they are greater than zero.

**A20.** *Page 13, line 413-5: The differences are again hardly visible, difficult to expect any effect on foehn, and thus to resolve if pure dynamical or thermodynamical influence*

**RESPONSE:**

We partly agree that the interpretation might be a bit superficial. We will weaken the statements a bit and formulate the foehn connection as a speculation in the revised version. Still, we would like to emphasise that the wind changes are statistical significant (Please refer to the minor comment A9). To clarify this statement, we will reformulate these lines in the revised version of the manuscript as follows:

"...The drier conditions at the southern face of the Alps may be explained by an enhanced Foehn effect due to the slightly clockwise turned winds (statistical significant). Thus, dynamical processes could also play a role to explain the summer precipitation changes between MIS4 and LGM."

**A21.** *Page 27, Fig. 4 and further: order of the periods in legend might be the same as for winds columns. In the caption missing explanation of the profile lines (temperature solid, dew point dashed) and correctly it should be . . . saturated mixing ratio increases . . .*

**RESPONSE:**

In the revised version of the manuscript, we will change the order of the periods according to the wind profiles. Also, we will add the explanation of the profile lines and change "mixing ratio" to "saturated mixing ratio".

Once again, we would like to thank the referee for the time invested in reviewing our manuscript so carefully. We look forward to meeting her/his expectations.

Best regards,

Patricio Velasquez (on behalf of the author team)

**References**

CORDEX: CORDEX – Coordinated Regional Climate Downscaling Experiment, URL `https://www.cordex.org/`, 2019.

Gómez-Navarro, J. J., Bothe, O., Wagner, S., Zorita, E., Werner, J. P., Luterbacher, J., Raible, C. C., and Montávez, J. P.: A regional climate palaeosimulation for Europe in the period 1500–1990 – Part 2: short-comings and strengths of models and reconstructions, Climate of the Past, 11, 1077–1095, https://doi.org/10.5194/cp-11-1077-2015, 2015.

Gómez-Navarro, J. J., Raible, C. C., Bozhinova, D., Martius, O., García Valero, J. A., and Montávez, J. P.: A new region-aware bias-correction method for simulated precipitation in areas of complex orography, Geoscientific Model Development, 11, 2231–2247, https://doi.org/10.5194/gmd-11-2231-2018, 2018.

González-Rojí, S. J., Sáenz, J., Ibarra-Berastegi, G., and Díaz de Argandoña, J.: Moisture balance over the Iberian Peninsula according to a regional climate model: the impact of 3DVAR data assimilation, Journal of Geophysical Research: Atmospheres, 123, 708–729, https://doi.org/10.1002/2017JD027511, 2018.

Hofer, D., Raible, C. C., Dehnert, A., and Kuhlemann, J.: The impact of different glacial boundary conditions on atmospheric dynamics and precipitation in the North Atlantic region, Climate of the Past, 8, 935–949, https://doi.org/10.5194/cp-8-935-2012, 2012a.

Hofer, D., Raible, C. C., Merz, N., Dehnert, A., and Kuhlemann, J.: Simulated winter circulation types in the North Atlantic and European region for preindustrial and glacial conditions: glacial circulation types, Geophysical Research Letters, 39, L15 805, https://doi.org/10.1029/2012GL052296, 2012b.

Landais, A., Masson-Delmotte, V., Capron, E., Langebroek, P. M., Bakker, P., Stone, E. J., Merz, N., Raible, C. C., Fischer, H., Orsi, A., Prié, F., Vinther, B., and Dahl-Jensen, D.: How warm was Greenland during the last interglacial period?, Climate of the Past, 12, 1933–1948, https://doi.org/10.5194/cp-12-1933-2016, 2016.

Merz, N., Raible, C. C., Fischer, H., Varma, V., Prange, M., and Stocker, T. F.: Greenland accumulation and its connection to the large-scale atmospheric circulation in ERA-Interim and paleoclimate simulations, Climate of the Past, 9, 2433–2450, https://doi.org/10.5194/cp-9-2433-2013, 2013.

Merz, N., Born, A., Raible, C. C., Fischer, H., and Stocker, T. F.: Dependence of Eemian Greenland temperature reconstructions on the ice sheet topography, Climate of the Past, 10, 1221–1238, https://doi.org/10.5194/cp-10-1221-2014, 2014a.

Merz, N., Gfeller, G., Born, A., Raible, C. C., Stocker, T. F., and Fischer, H.: Influence of ice sheet topography on Greenland precipitation during the Eemian interglacial, Journal of Geophysical Research: Atmospheres, 119, 10,749–10,768, https://doi.org/10.1002/2014JD021940, 2014b.

Merz, N., Raible, C. C., and Woollings, T.: North Atlantic eddy-driven jet in interglacial and glacial winter climates, Journal of Climate, 28, 3977–3997, https://doi.org/10.1175/JCLI-D-14-00525.1, 2015.

Merz, N., Born, A., Raible, C. C., and Stocker, T. F.: Warm Greenland during the last interglacial: the role of regional changes in sea ice cover, Climate of the Past, 12, 2011–2031, https://doi.org/10.5194/cp-12-2011-2016, 2016.

Rozsa, S.: Estimation of integrated water vapour from GPS observations using local models in Hungary, 136, 823, https://doi.org/10.1007/978-3-642-20338-1_03, 2012.

Velasquez, P., Messmer, M., and Raible, C. C.: A new bias-correction method for precipitation over complex terrain suitable for different climate states: a case study using WRF (version 3.8.1), Geoscientific Model Development, 13, 5007–5027, https://doi.org/10.5194/gmd-13-5007-2020, 2020.

Velasquez, P., Kaplan, J. O., Messmer, M., Ludwig, P., and Raible, C. C.: The role of land cover in the climate of glacial Europe, Climate of the Past, 17, 1161–1180, https://doi.org/10.5194/cp-17-1161-2021, 2021.

---

## Author Response (AR1)

**Final Response to Referee #1**

We thank the referee Maria Fernanda Sanchez Goñi for the careful and thorough reading of our manuscript. The comments have been carefully considered and responded. Please find below our response to each comment.

**General comments**

**1.    *The manuscript submitted by Velasquez et al. to the Climate of the Past presents a combination of global and regional model simulations to test the sensitivity of the glacial Alpine hydro-climate to northern hemisphere, Laurentidae and Fennoscandian, and local ice-sheet changes during the Last Glacial Maximum (LGM) and Marine Isotope Stage (MIS) 4. For the LGM, they find that thickening of the northern hemisphere ice-sheets, mainly the Laurentidae ice caps, and local ice-sheet topography generally lead to increase in winter precipitation and decrease in summer rainfall, both enhancing glacial conditions. In winter, dynamics processes related to the intensity and position of the Alpine winds explain the moistening in the southern part of the Alps while the simulated summer drying all over the Alps is related to thermodynamic processes, i.e. colder temperatures. In contrast, Fennoscandian ice-sheet changes have a negligible impact on the Alpine hydro-climate. For MIS 4, marked by lower global ice volume than the LGM, Velasquez et al. find wetter climate in the Alps attributed to thermodynamic processes, i.e. warmer temperatures. This manuscript is clearly written and convincing for the LGM. In contrast, I have several caveats related to the MIS 4 model results and comparison with the regional (western European) climate at that time when compared to that of the LGM (see below). Overall, this work deserves publication in CP after the authors address the comments that I have listed below. I am not a modelling expert, and I will only comment on the data discussed in this work.***

**RESPONSE:**

We thank you for your detailed comments. We have taken care of the concerns related to analysis of the MIS4 climate and the comparison with the LGM in the following responses and in the revised manuscript.

**Major comments**

**2.    *Lines 40-50: It would be relevant to cite the paper by Harrison and Digerfeldt (1993, Quaternary Science Reviews), one of the first paper showing that southern Europe was wet during the LGM (centered at 21 ka) based on the high water levels recorded in several lakes around the Mediterranean region. Iberian margin pollen records also provide evidence that the LGM in southern Europe was wetter than the Heinrich Stadial (HS) 2 and HS 1 bracketing it (Naughton et al., 2007, Marine Micropaleontology ; Turon et al., 2003, Quaternary Research).***

**RESPONSE:**

We agree that some pollen-based reconstructions show a wetter southern Europe compared to present day. Therefore, we reformulated these lines (lines 40-50) and added this information in the revised version of the manuscript. Additionally, we included other studies that highlight the wetter conditions in southern Europe compared to other periods, e.g. Heinrich Stadials and Marine Isotope Stages. The changes have been done as follows:

"...The same data are also used to reconstruct the hydro-climatic response over Europe at the LGM mainly showing drier conditions over Europe (reduction in precipitation of around 200 mm year$^{-1}$) compared to PD (Wu et al., 2007; Bartlein et al., 2011). Roucoux et al. (2005) indicated that the LGM was not necessarily a dry period everywhere in Europe. For instance, Turon et al. (2003); Roucoux et al. (2005); Naughton et al. (2007) and Ludwig et al. (2018) suggested that southern Europe was wetter compared to the rest of Europe and also to adjacent periods, i.e. the Marine Isotope Stage 3 (Voelker et al., 1997; van Kreveld et al., 2000) and the Heinrich events 1 and 2 (Sanchez Goñi and Harrison, 2010; Álvarez-Solas et al., 2011; Stanford et al., 2011). Compared to PD, many studies suggest that the wetter conditions in southern Europe can be explained by a southward shift in the North Atlantic storm track during the LGM (e.g. Hofer et al., 2012a; Luetscher et al., 2015; Merz et al., 2015; Ludwig et al., 2016; Wang et al., 2018; Raible et al., 2020; Lofverstrom, 2020). This southward shift is in line with other climate reconstructions that suggest circulation-induced changes in the moisture transport. For instance, Harrison and Digerfeldt (1993) found very different patterns of lake-level changes across Europe suggesting major changes in European atmospheric circulation patterns. Other climate reconstructions also suggest circulation-induced changes in the moisture transport (Florineth and Schlüchter, 2000). In this case, the atmospheric circulation is..."

**3.      Lines 50-54 and lines 407-415: To support the idea that MIS 4 was warmer and wetter than the LGM, Velasquez et al. only refer to global studies (Eggleston et al., 2016), Australasian records (De Deckker et al., 2019 ; Newham et al., 2017) and model simulations for the North Atlantic and Greenland climate (Hofer et al., 2012; Merz et al. papers) that cannot be used to account for the climate in Europe at that time and, particularly, at 65 ka, the date chosen for their simulations. This date is concomitant with the maximum of global ice volume during MIS 4 (Waelbroeck et al., 2002, Quaternary Science Reviews), coincides, within the chronological uncertainties, with Greenland Interstadial 18 and precedes the massive iceberg discharges in the North Atlantic leading to the HS 6, 64-60 ka (Sanchez Goñi et al., 2013, Nature Geoscience, Figure S3 of the supplementary information).**

**To realistically compared the recorded and simulated climate in Europe at 65 ka, the authors should discussed their wind field and climate reconstructions in the context of the climate prevailing in the western European margin and the adjacent landmasses during this period, climate that is mainly controlled by the westerlies during winter. The work by Sanchez Goñi et al. (2013, Nature Geoscience, Figure 2 and Figure S3 of the supplementary information) zooms in on MIS 4, and shows relatively wet and warm atmospheric conditions at 65 ka, based on the increase of heathlands and pine forest, contemporaneous with foraminifera-based warm summer sea surface temperatures in the western European margin, reaching 15°C in the Bay of Biscay and the SW Iberian margin and 10°C in the NW Iberian margin. However and in contrast with the authors' idea that the LGM was colder than MIS 4 in the European margin, higher sea surface temperatures in the Bay of Biscay (Sanchez Goñi, 2020, Evolutionary Human Sciences, Figure 2) and in NW and SW Iberia (Sanchez Goñi et al., 2008, Quaternary Science Reviews, Figures 3 and 4) are recorded during the LGM compared to MIS 4. Both periods are characterised by low and similar temperate forest abundance and similar heathlands development suggesting that MIS 4 was not warmer and wetter compared to the LGM.**

**RESPONSE:**

We agree that the description of MIS4 is too short and the citations do not focus on the European climate. We thank the referee for highlighting these studies. In the revised manuscript, we further introduced the MIS4 climate and briefly discussed its uncertainties. Therefore, we included the studies of Sánchez Goñi et al. (2008) and Sánchez Goñi (2020) and we have also reformulated these lines and added more information as follows:

Lines 50-54:

"... The MIS4 climate is less understood compared to the LGM as proxy data availability is further reduced compared to the LGM. Available paleoclimate reconstructions characterise MIS4 to be warmer than the LGM on a global scale (e.g. Eggleston et al., 2016; Newnham et al., 2017; De Deckker et al., 2019) with a global sea level drop of roughly 80 m compared to PD (e.g. Cutler et al., 2003; Siddall et al., 2008, 2010; De Deckker et al., 2019). Focusing on Europe, MIS4 shows relatively wet and warmer conditions at 65 ka compared to the period 85-50 ka (Sánchez Goñi et al., 2013). Still, Sánchez Goñi et al. (2008) and Sánchez Goñi (2020) found drier and colder conditions around the Iberian Peninsula during MIS4 compared to the LGM. Their findings suggest, similar to the LGM period, that the MIS4 climate was not necessarily homogeneously wetter and warmer across Europe."

Lines 407-415:

"...The MIS4$_{LGM}$ climate shows enhanced winter precipitation compared to the LGM. The reason is that the MIS4 climate state is generally warmer (Hofer et al., 2012a,b; Merz et al., 2013, 2014a,b, 2015, 2016) and thus more moisture is globally available. Wind changes do not contribute to these wetter conditions in the Alpine region as they become weaker and therefore reduce the orographically forced uplifts, which also suggests an overall reduction of the moisture transport to the Alps. Thus, we interpret the winter changes between MIS4 and LGM to be purely thermodynamically driven (Clausius–Clapeyron equation). In summer, MIS4$_{LGM}$ shows slightly wetter conditions at the northern face and drier conditions at the southern side of the Alps. The northern wetter conditions are induced by an increase in the tropospheric vertical wind shear enhancing convection processes. The drier conditions at the southern face of the Alps may be explained by an enhanced Foehn effect due to the slightly clockwise turned winds (statistically significant). Thus, dynamical processes could also play a role to explain the summer precipitation changes between MIS4 and LGM."

**4.    Line 395-406: The authors should add in the revised version of the manuscript the new evidence from a cryogenic carbonate record in the Alps (Spötl et al., 2021, Nature Comm.) showing heavy snowfall during autumn and early winter during the LGM. These results combined with thermal modelling, provide compelling evidence that the LGM glacier advance in the Alps was fuelled by intensive snowfall late in the year, likely sourced from the Mediterranean Sea.**

**RESPONSE:**

We appreciate the referee to bring to our attention this study. We have included this new evidence in the conclusions of the revised version.

**5.    Lines 436-440: The authors should delete the references of Finlayson et al., 2004, 2006 and 2008 and that of Burke et al., 2014 and Baena Preysler et al., 2019. These works do not refer to the Alpine**

*regions and, therefore, they are not relevant for this study.*

**RESPONSE:**

We have deleted these references in the revised version of the manuscript.

**Minor comments**

**m1.** *Line 72: Please add Regional Climate Models to explain RCM.*

**RESPONSE:**

We have added it in the revised version of the manuscript.

**m2.** *Line 158: Please replace « «eighth experiment » with « eighth experiments ».*

**RESPONSE:**

We cannot understand your suggestion as we refer to a single experiment in this sentence, i.e. the experiment number eight. To clarify any misunderstanding, we have reformulated this line in the revised manuscript as follows:

"...In the experiment number eight, we investigate..."

**m3.** *Line 255: Please replace « associated to » with « associated with ».*

**RESPONSE:**

We have replaced it in the revised version of the manuscript.

We would like to thank the referee Maria Fernanda Sanchez Goñi for the time invested in reviewing the manuscript so carefully. We are looking forward to meeting the referee's expectations.

Best regards,

Patricio Velasquez (on behalf of the author team)

**Final Response to Referee #2**

We appreciate the time the referee has invested in reading the manuscript in such a careful and thorough manner. The comments have been carefully considered and responded. Please find below our response to each comment.

**General comments**

**1.** *An interesting study investigating the sensitivity of the glacial Alpine hydro-climate to northern hemispheric and local ice-sheet changes, using the chain of GCM-RCM simulations to perform comparison and sensitivity analysis for two glacial periods, the LGM and MIS4, with the different ice-sheet thickness (in different glacial regions) effects on the Alpine hydro-climate conditions. The results are analyzed in very much detail, although in some cases the effects are rather small and with respect to the simulation concept, I would not be so sure that the conclusions are so firm (see some further more detailed comments).*

**RESPONSE:**

We thank you for your detailed comments. We have taken care of the concerns related to the conclusions in the following responses and in the revised manuscript.

**Major comments**

**2.** *First, to make these more strong at least some small ensemble (a few models) should be employed to see the robustness of the results.*

**RESPONSE:**

We agree that an ensemble can increase the robustness of the results, especially when analysing the climate simulations separately. However, we think that this suggestion is beyond the scope of our manuscript as this study focuses on topography-related sensitive experiments rather than assessing the uncertainties of the climate simulations. Moreover, we would like to mention that each regional climate simulation causes very high computational costs, e.g. calculation time and storage and our current systems at use do not allow for further simulations. Note further that we would also need to perform global model simulations. If the referee suggests to use different model chains, i.e., another GCM and another RCM, then this is certainly beyond the capacity of the group as this would imply gaining knowledge and experience with these models. Even modelling centres only use and maintain one model. Our guess is that the referee has a community effort in mind, so that a small ensemble would rather orientate within the modelling community, e.g. CORDEX (CORDEX, 2019). Clearly, we agree that this would be beneficial, but to our knowledge regional modelling in the paleoclimate perspective is rather new. We know only two other groups working on paleoclimate issues

and none has used convection permitting scales (except our group). Therefore, we see this study as a starting point and the robustness is given by statistical tests. We have been more specific in the conclusions that the results may depend on the model chain used. Thus, we added the following line at the end of the conclusions:

"Additionally, since the results of this study may depend on the chosen global and regional climate model, future modelling efforts are needed to perform more regional paleoclimate simulations, especially using different models to develop a model ensemble. This ensemble would allow to better assess the uncertainties of the simulated glacial climates."

**3.** *Second, the chain of the model domains is a bit strange. In my opinion, the innermost domain is too small to develop properly the circulation in the vicinity of the Alpine region and due to its location at the edge of further domain boundaries from all the three sides, the discussion of the results of simulation on the north-west and south-east sites are not equally valid. What comes from the south is more or less based on the 18 km resolution domain as with respect to the proximity to the other domains edges there is no enough room and time to properly develop in CP resolution. This might be of importance as there is a significant change of land-use in Adriatic till this border, as well as with respect to the shift of polar front under glacial conditions. I understand the limitations of these extensive and demanding simulations, however, these aspects should appear some way in the presentation and discussion of the results, with the limitations clearly declared and possible uncertainties pointed out.*

**RESPONSE:**

We appreciate that the referee brought to our attention that the model setup is still a bit unclear. We would like to mention that there is a relaxation zone that refers to the lateral areas of the domain where the WRF model is nudged or relaxed towards the larger-scale input data, i.e. the lateral boundary conditions. We used a relaxation zone of 5 grid points (10 km in the innermost domain D4) at each edge which is discarded from the analysis, i.e. the figures in manuscript do not show the relaxation zone. Without this relaxation zone in the innermost domain, there are still around 60, 80, 120 and 80 km distance (30, 40, 60 and 40 grid points, respectively) from the first elevated areas (e.g. above 1500 m a.s.l.) to the western, northern, eastern and southern edges of the domain, respectively. Furthermore, the domain D3 in 6 km resolution also uses a 5 grid point relaxation zone and is already convection permitting. Checking the simulations we find convection-type structures emerging also from the south, which where also found in simulations with the same setup but driven by reanalysis data (Gómez-Navarro et al., 2015, 2018). Still, we agree that a wider region would be beneficial. In the revised manuscript, we have clarified this in the method section and we will also mention it in the conclusion part as follows:

Model section: Page 4 , line 124

"...two domains. Also, we use a relaxation zone of five grid points (e.g. this means 10 km in the innermost domain) at each edge in each domain, which is not included in the analysis. WRF uses..."

Conclusion: Last paragraph

"Still, using a larger innermost domain would be beneficial in a future work to enhance the model-related development of the atmospheric circulation around the Alps."

**4.**     *Further, concerning the discussion of the relative humidity, I would like to stress the dependence of it on the actual temperature itself as well, which can be quite significant when comparing the PD and LGM, see below in specific comments, but please check throughout the paper. The same distance between the dew point and actual temperatures under these different conditions of PD and LGM will not mean the same relative humidity.*

**RESPONSE:**

We fully agree that the relative humidity also depends on the air temperature which is strongly different between PD and LGM. Also, we think that the distance between the dew point and air temperature would poorly represent the relative humidity on a linearly scaled diagram, not even as a qualitative representation. However, we believe that the logarithmic scale of the Skew–T diagram allows to interpret the distance between the dew-point temperature and air temperature as a qualitative representation of the relative humidity. We have taken care of this concern in the following responses and in the revised version of the manuscript.

**5.**     *Moreover, concerning the mixing ratios in the diagram, that means saturated mixing ratio under the actual atmospheric conditions, which is not saying too much about the precipitable water, it depends on the relative humidity. Connection to Clausius-Clapeyron equation makes more sense in the discussion of extreme precipitation, where really different temperature of the atmosphere with different maxima of potential mixing ratios results in different amount of precipitable water. However, in the results presented the relative humidity looks to be rather lower.*

**RESPONSE:**

We have changed the name of the mixing ratio lines (page 6 lines 178) to saturated mixing ratio lines in the revised manuscript. Additionally, we have modified the following lines (page 6 lines 185-186):

"...dew-point temperatures (dashed lines). The latter simply indicate temperatures at which the air becomes saturated and is used to deduce the mixing ratio with height, i.e., the amount of water vapour in the air where the dew point temperature line crosses the mixing ratio line."

to

"...dew-point temperatures (dashed lines). The air temperatures are used to deduce the mixing ratios with height, i.e. the amount of water vapour in the air with height, whose values are obtained from the saturated mixing ratio lines when they are crossed by the temperature vertical profiles. The dew-point temperatures indicate the temperatures at which the air becomes saturated."

Regarding the precipitable water (PW), also called integrated water vapour (IWV), we agree that the PW is not directly represented by the mixing ratio at one pressure level. Note that the value of the mixing ratio is obtained from the saturated mixing ratio line when it is crossed by the air temperature vertical profile. The PW represents the amount of water vapour of an atmospheric column expressed as the depth of water if that vapour were condensed, i.e. water vapour available for precipitation. The PW value is calculated by vertically integrating the water vapour mixing ratios across height and it does not depend on the temperature (Eq. 1; e.g. Rozsa, 2012; González-Rojí et al., 2018). These values are shown at the top of each panel of figure 4 and 8 in the manuscript.

$$PW = IWV = \frac{1}{g} \int_{p_{sfc}}^{p_{top}} q \, dp, \tag{1}$$

where $g$ is the gravitational acceleration, $q$ the vapour mixing ratios, $p_{top}$ and $p_{sfc}$ the pressure at the uppermost and lowermost model level, respectively. We will clarify this in the method and result section of the revised version of the manuscript.

To clarify the interpretation of PW, we have added more information on page 6 in line 193 as follows:

"...energy (CAPE). PW represents the amount of water vapour of an atmospheric column expressed as the depth of water if that vapour were condensed, i.e. water vapour available for precipitation. The PW value is calculated by vertically integrating the water vapour mixing ratios across height and it does not depend on the temperature (e.g. Rozsa, 2012; González-Rojí et al., 2018). Cape quantitatively..."

**6.    *A formal comment concerns the rotation of the wind (appearing throughout the paper). I would recommend using the term turning of the wind, which is commonly used when describing the changes of wind direction with height, i.e. wind turns clockwise or anticlockwise. Similarly, something can cause to turn the wind to some direction, e.g. changes of ice-sheet heights, LGM conditions etc.***

**RESPONSE:**

We appreciate this suggestion. We have changed the term "rotation" to "turning" in the revised manuscript.

**Specific comments**

**A1.    *Page 1, line 7: explained***

**RESPONSE:**

It has been changed.

**A2.    *Page 4, line 126-9: Actually, this simulation strategy is not too rigorous comparing what is commonly required from RCM simulations***

**RESPONSE:**

We partly agree with the referee. A better simulation strategy would generally be when an RCM runs for 30 years in a single piece with a spin-up period longer than 2 months. However, we think that this single simulation strategy hardly influences our conclusions, especially as our analyses are completely based on mean values, i.e. the climatology. We would like to highlight the reason for choosing our strategy in the following:

Splitting up the simulations is explained by the time-consuming setup to run a simulation over the Alps at 2 km resolution. Namely, three model years are equivalent to one month in real time, which means

that a 21-years simulation in a single piece would have taken at least seven months in real time without any interruption. Each regional climate simulation is forced under perpetual conditions, i.e. constant climate conditions, which allows to split it into 3-years simulations that represent the same climate state. For example, a 21-years simulation is performed under perpetual 1990 conditions. To that end, the 21-years simulation is split into seven individual 3-years simulations which all represent the same 1990 climate conditions. We have briefly mentioned the common strategy and reformulated these lines in the revised version of the manuscript as follows:

"...simulation segments, respectively. Note that we split the simulations to efficiently use the available computer facilities (similar to accompanying studies such as Velasquez et al., 2020, 2021), even though regional climate simulations would commonly be performed in one single simulation. For each segment, a 2-month..."

**A3.** *Page 6, line 171: not clear how 30 annual mean samples can be selected from 21 or even 12 years simulations?*

**RESPONSE:**

We thank the referee for this comment. The number "30" is a mistake. It has been deleted in the revised version of the manuscript.

**A4.** *Page 6, line 188-9 the relative humidity still depends on the actual temperature as well, the same difference between the actual and dew point temperature does not imply the same relative humidity, especially under quite different temperatures like for PD and LGM*

**RESPONSE:**

We fully agree that relative humidity (RH) depends on the actual air temperature as well. This dependency is implicitly included in the Skew–T diagram and therefore the values of RH can be qualitatively evaluated by the distance between the temperature profiles, i.e. the distance between the dew-point temperature and air temperature. We have clarified this in the revised version of the manuscript as follows:

Lines 186-189:

"...Both temperatures are used to investigate the relative humidity, i.e. the level of saturation at a certain pressure and temperature. Note the Skew–T diagram implicitly includes the dependency of the relative humidity on temperature by the logarithmic scale of the saturated mixing ratio lines. Therefore, the relative humidity can be obtained by qualitatively estimating the distance between both temperature profiles. A short distance indicates a high relative humidity and, inversely, a large distance a low relative humidity."

**A5.** *Page 8, line 230: In fact, mixing ratios are not shown, in the diagram, there are saturated mixing ratios for the given conditions provided, precipitable water depends on actual relative humidity*

**RESPONSE:**

We have changed the name of these lines in the revised manuscript. Also, we clarified the concern about the relative humidity and precipitable water in a previous answer. Please refer to our response to the

fifth major comment.

**A6.** *Page 8, line 231-2: relative humidity is not the same with the same difference between the dew point temperature and actual temperature, it depends on temperature itself as well (see above)*

RESPONSE:

We clarified this concern in a previous answer. Please refer to our response to the fifth major point.

**A7.** *Page 8, line 244: I do see boundary layer in PD as well, of course, clearly, with a less stable lapse rate due to surface warming in summer*

RESPONSE:

We agree that there is a visible boundary layer in PD as well with less stable lapse rate due to surface warming in summer. We have modified this line as follows:

"...layer under PD and LGM conditions, particularly distinct during the LGM."

**A8.** *Page 8, line 249: Actually, for the wind field (circulation) analysis rather larger scale (domain) should be shown with pressure fields changes, which will be probably a stronger driver of the circulation. The alpine effect can be seen just in the close proximity, where even in 700 hPa level Alps can create barrier with different height during PD and LGM (MIS4) inducing either overflow or flow around some parts, which can be well resolved in the highest resolution (in connection to stability as well)*

RESPONSE:

We would like to mention that the PD and LGM CCSM4 simulations have been analysed over Europe in a variety of studies, including additional simulations for other glacial and interglacial states (e.g. Hofer et al., 2012a,b; Merz et al., 2013, 2014b,a, 2015, 2016; Landais et al., 2016). These studies particularly focused on changes in the atmospheric circulation during glacial times. Compared to PD conditions, the LGM simulation reveals a clear southward shift and a more zonal orientation of the storm track over the North Atlantic and substantial changes in the weather patterns (Hofer et al., 2012b,a). These changes are able to explain precipitation anomalies over Europe, especially over the Iberian Peninsula and the western part of the Mediterranean Sea.

We have analysed the larger-scale atmospheric circulation in our simulations using wind vectors at 700 hPa for the second domain (over Europe at 18 km resolution). Compared to PD, winter winds are turned anticlockwise (more zonal) and generally intensified during the LGM, which is mainly attributed to significant changes in both wind components (Fig. 1a). We also observe that the winter jet stream is shifted southward during the LGM. In summer, winds overall show a north-south turning pattern with the axis at around 48 °N during the LGM (Fig. 1c): anticlockwise and clockwise turning in the northern and southern part of the domain, respectively. This north-south turning pattern is mostly associated to either significant changes in the zonal component (U) only or both wind components (i.e. areas in blue or red, respectively; Fig. 1c). Compared to the LGM, $MIS4_{LGM}$ winds at 700 hPa become weaker with an almost absent turning in winter (Fig. 1b), whereas they are slightly stronger with a minor clockwise turning in summer (Fig.

1d). Furthermore, we observe that winds at 700 hPa are generally intensified and turned anticlockwise with increasing the ice-sheet thickness (Fig. 2). Namely, increasing the thickness of the northern-hemispheric ice sheet leads to generally more zonal winter winds in the western part of the domain, which suggests that the flow has a warmer source and therefore more water vapour available. This would confirm the wetter conditions in winter (comparing Fig. 5b and c of the revised manuscript). Note that winds mainly show very weak differences for the sensitivity of the FIS over Europe (therefore not shown). Same results were already found in the previous studies over Europe (above mentioned). This indicates that there is a strong correlation of the larger-scale atmospheric circulation between the driving CCSM4 and the WRF simulations. Therefore, we believe that including a in-detail analysis would be a repetition of these previous studies.

Nevertheless, we have included a brief analysis of the larger-scale atmospheric circulation in section 4.1 (sixth paragraph) of the revised version of the manuscript as follows:

"...To gain further insights in the advection of moisture, we first focus on the larger-scale mid-atmospheric circulation for the second domain (over Europe at 18 km resolution). The comparison to PD conditions shows that the climatological mid-atmospheric mean flow turns anticlockwise with stronger wind speeds, which indicates a more zonal, intensified and southward shifted winter jet stream during the LGM. Same results were already found in previous studies over Europe, which analysed the CCSM4 simulations that drive our WRF simulations (Hofer et al., 2012a,b). This indicates that there is a strong correlation of the larger-atmospheric circulation between the driving CCSM4 and the WRF simulations. Note that these CCSM4 simulations and their underlying atmospheric circulation have been already analysed over Europe in a variety of studies (e.g. Hofer et al., 2012a,b; Merz et al., 2013, 2014b,a, 2015, 2016; Landais et al., 2016); therefore please refer to these studies for a in-detail analysis of the European atmospheric circulation. Secondly, we exhibit the wind vectors at 700..."

[Figure]

Figure 1: Climatological mean wind vectors over Europe for (a and b) DJF and (c and d) JJA: (a and c) black and green wind vectors correspond to PD and LGM, respectively, (b and d) black and green wind vectors correspond to LGM and MIS4, respectively. Red shading illustrates statistically significant differences in zonal (U) and meridional (V) wind components with a significance level of 0.05 (two-tailed bootstrapping technique), blue and grey shading indicate significance either in the U or V wind component, respectively. Inner box represents the innermost domain (at 2 km), i.e. the Alpine region. Please note that the reference wind vectors differ for DJF and JJA.

[Figure]

Figure 2: Climatological mean wind vectors over Europe. (a) represents wind vectors for DJF, black and green vectors correspond to MIS4$_{LGM}$ and MIS4$_{LGM66}$, respectively, (b) as (a) but green vectors correspond to MIS4$_{LGM125}$. (c) and (d) as (a) and (b) but for JJA. Red shading indicates statistically significant differences in zonal (U) and meridional (V) wind components at a significance level of 0.05 (two-tailed bootstrapping technique), blue and grey shading as the red one but only in U and V wind components, respectively. The box represents the innermost domain (with 2 km horizontal resolution), i.e. the Alpine region. Please note that the reference wind vectors differ for DJF and JJA.

**A9.** *Page 9, line 279: Actually, I do not see so much significant difference (despite formal statistical significance) to discuss here except the cases of the proximity of Alpine ridge, where the differences can be due to different heights of the terrain (glaciers), as mentioned above*

**RESPONSE:**

The referee refers to Fig 5d. There, the Alpine ice sheet is not changed so the changes are not due to topographic differences. We agree that the wind vectors do not suggest the similarly strong changes in Fig 5d as in Fig 5a. Still, the shading shows the statistical significance of the changes. We have added few information in the previous lines to better clarify the topographic differences as follows:

"...Therefore, we assess the wind vectors at 700 hPa for the innermost domain (over the Alpine region at 2 km resolution). Note that the LGM and $MIS4_{LGM}$ simulations use the same topography, i.e. the LGM topography. Figure 5d..."

Maybe the visualisation problem has been due to the chosen features of the wind vectors; therefore, we have redesigned the figure to better represent the wind vectors.

**A10.** *Page 10. Line 319: it is rather tiny*

**RESPONSE:**

As answered in previous response, we have faced this concern in the analysis of the larger-scale wind fields at 700 hPa in the revised version of the manuscript.

**A11.** *Page 10, line 321: This will be really negligible, especially with respect to the relative humidity, which will be rather low. By the way, again, the green dashed lines represent saturated mixing ratios, not actual mixing ratios.*

**RESPONSE:**

As in response to the fifth major comment, we would like to mention that a mixing ratio is obtained from the saturated mixing ratio line when it is crossed by the vertical profile of the temperature. Regarding the definition of precipitable water, please refer to the same response (fifth major comment). We have reformulated this line and added additional information in the revised version of the manuscript as follows:

"...warmer atmosphere. Even though the Skew–T diagram indicates that the relative humidity is rather low, the warmer atmosphere results in a small increase of moisture availability in the middle-to-low atmosphere (water vapour). Note that the higher moisture availability is illustrated by the increase of the values of the mixing ratio, which are obtained from crossing the saturated mixing ratio lines with the vertical profile of the air temperature. This moisture increase is especially true for the central-southern region (site B; Fig. 8c) where there is more precipitable water (PW values at the top of Fig. 8a and c)."

**A12.** *Page 10, line 325: As above, saturated mixing ratios*

**RESPONSE:**

The name has been changed in the revised manuscript accordingly.

**A13.** *Page 10, line 328: glacial climate conditions in Alpine region*

**RESPONSE:**

We thank the referee. This has been changed in the revised version of the manuscript.

**A14.** *Page 11, line 335-6: Again, the changes are really very tiny, despite statistical significance*

**RESPONSE:**

We agree that the differences are visually very small. As mentioned in the response to the minor comment A9, we have faced this concern in the analysis of the larger-scale wind fields at 700 hPa in the revised version of the manuscript.

**A15.** *Page 11, line 354: Actually, it is difficult to see the significance in such a small region*

**RESPONSE:**

In the revised version of the manuscript, we have changed the visualisation of the significance to another drawing pattern.

**A16.** *Page 11, line 355-7: Actually, direct westerly inflow is more or less perpendicular to the barrier of the Alpine ridge in the western part, while in the eastern part it is rather parallel.*

**RESPONSE:**

We agree with the referee. In these lines, we referred the precipitation changes to the modification of ice-sheet thickness rather than to wind changes. To clarify this, we have added more information to these lines at the end of the following paragraph of the revised manuscript as follows:

"...Note that wind changes in winter cannot directly explain the stronger reduction of winter precipitation in the western part (see previous paragraph), which are caused by the reduction of the Alpine ice-sheet thickness. The reason is that we would expect wetter conditions in the western part since the winds slightly turn clockwise (more perpendicular to the Alps) and slightly stronger. Additionally, the further analysis of the larger-scale shows no differences (therefore not shown)."

**A17.** *Page 12, line 371-2: Actually, despite the formal statistical significance I do not see so big changes except in the close proximity of the mountain ridge (as noticed above), where the direct interaction of the*

*changed top of the barrier is evident and causing direct changes in flow patterns on that level.*

**RESPONSE:**

We agree with the referee. We have reformulated these lines in the revised manuscript as follows:

"...The LGM$_{\text{ALPSLESS}}$ winds are a bit stronger and slightly turned clockwise compared to LGM$_{\text{LGM}}$. The slight turning is associated with significant changes in V at the northern face of the Alps, in both wind components over the Alpine axis and in U in the south of the Alps (Fig. 11a)."

**A18.** *Page 12, line 374-7: However, a strong issue is how the model represents the transfer of precipitation from the place of creation downstream with the flow to the place it is considered as reaching the surface.*

**RESPONSE:**

Our guess is that the referee suggests to perform a back trajectory analysis to identify where the precipitation is coming from, e.g. North Atlantic, land surface over Europe (water recycling) and the Mediterranean. Clearly, this would be an interesting question, but we think that this is beyond the scope of this study but certainly provides a next step, which has been pointed out at the end of the manuscript.

**A19.** *Page 13, line 405-6: This would be nice shown on the analysis of convective precipitation*

**RESPONSE:**

We would like to mention that this analysis is not possible as our simulations are convection-permitting in the two innermost domains (6 and 2 km). This does not allow to separate the convective precipitation from the large-scale precipitation since the total precipitation is directly resolved in the model. CAPE is another quantitative measure that refers to convective activity. CAPE values are shown at the top of each panel on figure 4 and 8, when they are greater than zero.

**A20.** *Page 13, line 413-5: The differences are again hardly visible, difficult to expect any effect on foehn, and thus to resolve if pure dynamical or thermodynamical influence*

**RESPONSE:**

We partly agree that the interpretation might be a bit superficial. We have weakened the statements a bit and formulated the Foehn connection as a speculation in the revised version. Still, we would like to emphasise that the wind changes are statistical significant (please refer to the minor comment A9). To clarify this statement, we have reformulated these lines in the revised version of the manuscript as follows:

"...The drier conditions at the southern face of the Alps may be explained by an enhanced Foehn effect due to the slightly clockwise turned winds (statistically significant). Thus, dynamical processes could also play a role to explain the summer precipitation changes between MIS4 and LGM."

**A21.** *Page 27, Fig. 4 and further: order of the periods in legend might be the same as for winds columns. In the caption missing explanation of the profile lines (temperature solid, dew point dashed) and correctly it should be . . . saturated mixing ratio increases . . .*

**RESPONSE:**

In the revised version of the manuscript, we have changed the order of the periods according to the wind profiles. Also, we have added the explanation of the profile lines and changed "mixing ratio" to "saturated mixing ratio".

Once again, we would like to thank the referee for the time invested in reviewing our manuscript so carefully. We look forward to meeting the referee's expectations.

Best regards,

Patricio Velasquez (on behalf of the author team)

[revised manuscript text omitted]

---

## Author Response (AR2)

**Response to Editor's comments**

We thank the Editor Dr. Qiuzhen Yin for the time invested in reading the manuscript and the reviews in such a careful and thorough manner. The comments have been carefully considered and responded. Please find below our response to each comment.

**General comments**

**1.    As you will see, Reviewer 2 still has some concerns and comments. The major concern of Reviewer 2 is on the uncertainty of using the single model simulation. In the meantime, the reviewer also acknowledges the potential cost/difficulty to solve the methodology issues raised. As a compromise, the reviewer suggests to provide enough warning about the reliability of the results and the uncertainty of the method used in both the methodology section and at the beginning of the conclusions. Please consider this suggestion as well as other comments of the reviewer in your next revision.**

**RESPONSE:**

We appreciate your detailed comments. We have taken care of the concerns of the Reviewer 2 in the responses to his comments and in the revised manuscript.

**Specific comments**

**2.    Add a "Data availability" section in your paper, which is requested by Climate of the Past**

**RESPONSE:**

We thank you for bringing it to our attention. This section has been included in the revised version of the manuscript.

**3.    As far as I can see, the astronomical parameters of Berger (1978, Journal of atmospheric Sciences) are used for the 1990, LGM and MIS4 simulations, so this original paper for the astronomical parameters should be cited. Please add Berger (1978) in your reference list**

**RESPONSE:**

We fully appreciate the reminder. We truly apologise for missing this part in the previous version of the manuscript. We reformulated the caption as follows:

"External forcing used in this study for 1990 CE, LGM and MIS4 conditions. The orbital parameters are calculated according to Berger (1978). Estimates for glacial levels of $CO_2$, $CH_4$ and $N_2O$ are obtained for LGM from the PMIP protocol (http://www-lsce.cea.fr/pmip2/) and for MIS4 from ice cores according to Schilt et al. (2010) and Bereiter et al. (2014). Note that the external forcing corresponds to the values of the driving the CCSM4 simulations (Hofer et al., 2012a,b)"

**4.**     *In Table 1, is "15.22" is typing error? Shouldn't it be 195.22?*

**RESPONSE:**

We have checked our calculations for the orbital parameters according to Berger (1978) and the value 15.22 is correct. This value was also used in the CCSM4 simulation that functions as initial and boundary conditions. To clarify this, we reformulated the caption, please refer to the previous response.

**5.**     *Regarding the experiments LGM(FIS50) and LGM(FIS150), the size of FIS is changed only in WRF but not in LGM CCSM4. What would be the impact of this inconsistency of ice sheet setup between the GCM and RCM on the results?*

**RESPONSE:**

We are aware of a potential effect due to the unchanged LGM ice sheet in the CCSM4 simulation. Nevertheless, we expect only a minor impact of this experimental design on the conclusion since we rather strongly modified the ice-sheet configuration in the WRF simulations. To assure that the reader is aware of a potential impact, we added further information in the section about the model setup (Sect. 2 of the revised manuscript) and in the discussion and conclusions (Sect. 5 of the revised manuscript).

In Sect. 2:

"...Note that this CCSM4 simulation uses 100 % LGM ice-sheet configurations..."

to

"...Note that this CCSM4 simulation uses 100 % LGM ice-sheet configurations for both sensitive simulations...."

In Sect. 5:

"...One potential reason of this weak precipitation response may also be the design of the Fennoscandian sensitivity experiment as the driving GCM has not experienced the changes of the FIS. However, we introduced rather strong ...".

to

"...One potential reason of this weak precipitation response may also be the design of the Fennoscandian sensitivity experiment as the driving GCM has not experienced the changes of the FIS, as the FIS was only modified in the RCM. However, we introduced rather strong ...".

**6.**     *As land cover could be critical for regional climate, the potential impact of using the LGM(LGM) land cover in the MIS4 experiments should be commented in the paper.*

**RESPONSE:**

We agree that the land cover plays an important role in realistically representing the climate as demonstrated by the accompanying study of Velasquez et al. (2021) and by other other studies (e.g. Kjellström et al., 2010; Strandberg et al., 2011; Ludwig et al., 2017). We had mentioned this in the introduction and we included additional information about the model setup in the section 2 of the revised version of the manuscript:

"...21 years using MIS4 conditions and using the $LGM_{LGM}$ land cover (described in Velasquez et al., 2021). Note further that the Alpine ice sheet is..."

to

"...21 years using MIS4 conditions and using the $LGM_{LGM}$ land cover (described in Velasquez et al., 2021). We use a $LGM_{LGM}$ land cover because it is the closest approach to a MIS4-like land cover as a gridded European MIS4 land cover has not been developed yet. Even though the $LGM_{LGM}$ land cover might influence the representation of the MIS4 climate, it provides the opportunity to gain insights into the effects of a different orbital and atmospheric forcing, i.e. by comparing $LGM_{LGM}$ to $MIS4_{LGM}$. Additionally, the Alpine ice sheet is..."

Once again, we would like to thank the editor Dr. Qiuzhen Yin. We are looking forward to meeting the editor's expectations.

Best regards,

Patricio Velasquez (on behalf of the author team)

**Response to referee #1**

We appreciate the time the reviewer invested in reading the manuscript in such a careful and thorough manner. We carefully considered and responded the comments. Please find below our response to each comment.

**General comments**

**1.** *Thanks to the authors for the explanations and answers to the comments. To be honest, I understand the argument on demanding simulations, on the other hand, to spend even a year with RCM simulations is not anything exceptional and I do not think it is correct to neglect some „standards" for dynamical downscaling, established within a long time of its development and used in the activities like CORDEX. Not considering the ensemble experiments, which really need the coordinated effort of many groups (already achieved to some extent in global scale within CMIP6 even for paleo, perhaps for regional-scale the opportunity could be raised within CORDEX – FPS), however, even for single modelling setup, this makes a sense as it can guarantee to some extent the reliability of the study. This is as my opinion the weakest point of presented contribution, especially this is i) the separation to shorter periods of the simulation, which do not allow to develop fully the own RCM climate, ii) very short spin-up, which perhaps can provide quasi-equilibrium in some short time scale in a specific season, but the question is if it is working properly in a full annual cycle, as the processes connected with deep layers are slow and delayed, and iii) the location of domains. Clearly, I understand it is impossible to solve these issues as this means to redo the study, and I do not like to waste the time the authors spend with careful analysis of the results and their interpretation, while I really appreciate the attempt to use this kind of tools in paleoclimate studies. However, I do not think that just a sentence at the end of the conclusions provides enough warning about the reliability of the results. In my opinion, this should be mentioned and elaborated a bit both in the methodology section as well as at the beginning of the conclusions. Significance based on statistical bootstrap technique can provide the information on the significance of differences, although sometimes really rather negligible, but not saying too much about overall robustness of the results, e.g. some results from CORDEX FPS LUCAS show big differences in RCM reaction to land cover changes, which might play a role even in this study.*

**RESPONSE:**

We thank you for your detailed comments. To provide enough warning about the reliability of the results, we included some information in the section about the method and the conclusion of the revised manuscript as follows:

At the end of the third paragraph of Sect. 2:

"...These 21 and 12 years are further split up into 7 and 4 individual 3-year simulation segments, respectively. Note that we split the simulations to efficiently use the available computer facilities similar to accompanying studies such as Velasquez et al. (2020, 2021), even though regional climate simulations would commonly be

performed in one single simulation. For each segment, a 2-month spin-up is needed in order to allow the land surface to come into quasi-equilibrium. Tests suggest that a 2-month spin-up is sufficient to obtain a quasi-equilibrium of the upper meter of the land surface (Velasquez et al., 2020, 2021).

to

"...These 21 and 12 years are further split up into 7 and 4 individual 3-year simulation segments, respectively. We split the simulations to efficiently use the available computer facilities (similar to accompanying studies such as Velasquez et al., 2020, 2021), even though regional climate simulations would commonly be performed in one single simulation. Note that all segments represent the same climate conditions, i.e. they are driven by the GCM using the same climate state. For each segment, a 2-month spin-up is needed in order to allow the land surface to come into quasi-equilibrium. To determine this spin-up, we analyse each segment and the results show that a 2-month spin-up is sufficient to obtain a quasi-equilibrium of the upper meter of the land surface in this study, i.e. no significant trend in the layers of the WRF land-surface scheme. Accompanying studies also suggest a 2-month spin-up (Velasquez et al., 2020, 2021). Note that the spin-up might vary if the segment starts in another season, which is not possible to test in this study due to the highly expensive model settings...

At the end of the first paragraph of Sect. 3:

"...The bootstrapping technique is applied at each grid point using as elements the annual mean values..."

to

"... The bootstrapping technique is applied at each grid point using as elements the annual mean values. Note that few results may be considered negligible, e.g. due to their very small differences, even though there is a statistical significance given by the bootstrapping technique..."

From fourth line of the third paragraph of Sect. 2:

"...The domains focus on the Alpine region; the outermost domain includes Europe and part of the North Atlantic to capture the influence of the North Atlantic Ocean and the FIS on the European climate (Fig. 1). Furthermore, we us..."

to

"...The domains focus on the Alpine region; the outermost domain includes Europe and part of the North Atlantic to capture the influence of the North Atlantic Ocean and the FIS on the European climate (Fig. 1). Note that the domain setup might influence the Alpine climate, which is not possible to test in this study due to the highly expensive model settings. Furthermore, we us..."

At the end of the first paragraph of conclusion section:

"...Note that the results might depend on the model setup, which is not possible to test in this study due to the highly expensive model settings..."

**2.** *Further, concerning the skew-T diagram, I am sorry, but it was a misunderstanding, as documented in the authors' response: Additionally, we have modified the following lines (page 6 lines 185-186):*

*"...dew-point temperatures (dashed lines). The latter simply indicate temperatures at which the air be-comes saturated and is used to deduce the mixing ratio with height, i.e., the amount of water vapour in the air where the dew point temperature line crosses the mixing ratio line." Which is correct, but was replaced by an incorrect sentence appearing then in other places of results discussion (see specific com-ments, actually, maybe not all found, please check throughout the paper). Indeed, real actual mixing ratio is given on crossing of dew point curve and saturation mixing ratio line. On the crossing of temperature and saturation mixing line, there is a saturated mixing ratio for a given temperature, which is not saying anything on real actual moisture availability.*

**RESPONSE:**

We changed it in the revised manuscript according to the referee's suggestion.

**Specific comments**

**3.** *Page 7, line 196: I would prefer the use of „saturation adiabatic lines", which is more correct*

**RESPONSE:**

We changed it in the revised manuscript according to the referee's suggestion.

**4.** *Page 7, line 197: Should be probably „air parcel"*

**RESPONSE:**

We changed it in the revised manuscript according to the referee's suggestion.

**5.** *Page 7, lines 202-204: The air temperatures are used to deduce the mixing ratios with height, i.e. the amount of water vapour in the air with height, whose values are obtained from the saturated mixing ratio lines when they are crossed by the temperature vertical profiles. - Actually, the real actual mixing ratios are saturated mixing ratios crossed by dew point lines.*

**RESPONSE:**

We reformulated these lines in the revised manuscript according to the referee's suggestion as follows:

"...the mixing ratios with height, i.e. the amount of water vapour in the air with height, whose values are obtained from the saturated mixing ratio lines when they are crossed by the temperature vertical profiles. The dew-point temperatures indicate the temperatures at which the air becomes saturated. Both temperatures are used to investigate..."

to

"...the mixing ratios with height. The real mixing ratios with height, i.e. the amount of water vapour, are obtained from the saturated mixing ratio lines when they are crossed by the dew-point temperature vertical profile. The dew-point temperatures indicate the temperatures at which the air becomes saturated. Both air and dew-point temperatures are used to investigate..."

**6.** *Page 7, line 221: „ ... for two glacial states the LGM and MIS4. „ Should be comma or colon in front of the LGM ... ?*

**RESPONSE:**

Yes, a colon was added in the revised manuscript.

**7.** *Page 8, line 253: The mixing ratios (the saturated mixing ratio lines crossed by the air temperature vertical profile) – Actually, the real actual mixing ratios are saturated mixing ratios crossed by dew point lines.*

**RESPONSE:**

We reformulated these lines in the revised manuscript according to the referee's suggestion.

**8.** *Page 10, line 301: The same as above*

**RESPONSE:**

We reformulated these lines in the revised manuscript according to the referee's suggestion.

**9.** *Page 10, lines 304-305: Hardly to see the difference. Actually, the angle to the adiabats matters and this remains basically the same despite there is slight parallel difference between the temperature curves*

**RESPONSE:**

We appreciate the referee for bringing this to our attention. Indeed, these lines mislead the interpretation of the figure. Therefore, we reformulated these lines in the revised manuscript as follows:

"...This reduction is more evident in the north-western region (site A), whereas the stability in the central-southern region (site B) is slightly reduced in the mid layer of the troposphere (between 800 and 400 hPa; Fig...."

to

"...This reduction is more evident during winter in both regions, i.e. north-western (site A, Fig. 4a) and central-southern (site B, Fig. 4c..."

**10.** *Page 10, line 317: How this is working? How that hard to see a change of wind direction can affect the available moisture via foehn effect, actually, the flow is coming over the Alps anyway.*

**RESPONSE:**

We agree that this is hard to see in the figure. Since we do not explicitly demonstrate the Foehn process, we weakened the statement and reformulated these lines as a speculation for the dryness in Fig. 3h as follows:

"...the northern part of the Alps, whereas the clockwise turning over the southern face of the Alps in MIS4 leads to reduced moisture availability as the flow dries out when crossing the Alps and reaching the Po valley (Foehn process)..."

to

"...northern part of the Alps. Over the southern face of the Alps, the slightly clockwise turning could lead to reduced moisture availability in MIS4 as the flow would tend to dry out when crossing the Alps and reaching the Po valley (assuming a Foehn process)..."

**11.** *Page 11, lines 343-344: Not exactly true for northern part of the Alpine region, I do not see linear response there.*

**RESPONSE:**

We apologise for this misunderstanding. In winter, the precipitation response is interpreted as linear with respect to the northern hemispheric ice-sheet thickness changes when comparing the difference between $MIS4_{LGM125}$ and $MIS4_{LGM}$ with the one between LGM and PD. We reformulated these lines in the revised manuscript as follows:

"...In winter, the difference pattern in precipitation between $MIS4_{LGM125}$ and $MIS4_{LGM}$ is similar to the one found between LGM and $MIS4_{LGM}$ with overall wetter conditions. Especially, we find significantly high precipitation intensities up to 3 mm day$^{-1}$ on the north western and southern regions of the domain (Fig. 6c). The northern face of the Alps shows a decrease in the precipitation intensities. Thus, we interpret the response of winter precipitation as linear with respect to the northern hemispheric ice-sheet thickness changes. In summer, ..."

to

"...In winter, the difference pattern in precipitation between $MIS4_{LGM125}$ and $MIS4_{LGM}$ is generally similar to the one found between LGM and PD. Especially, we find significantly high precipitation intensities up to 3 mm day$^{-1}$ on the north western and southern regions of the domain (Fig. 6c). The northern face of the Alps shows a decrease in the precipitation intensities. Thus, we interpret that the response of winter precipitation is linear with respect to the northern hemispheric ice-sheet thickness changes. In summer, ..."

**12.** *Page 11, lines 359-360: Again, actual mixing ratio is given by crossing of dew point curve and saturated mixing ratio line, actually, changes rather low*

**RESPONSE:**

We changed it according to referee's suggestion. Additionally, we reformulated this line to avoid any misleading in the revised manuscript as follows:

"...Even though the Skew-T diagram indicates that the relative humidity is rather low, the warmer atmosphere results in a small increase of moisture availability in the middle-to-low atmosphere (water vapour). Note that the higher moisture availability is illustrated by the increase of the values of the mixing ratio, which are obtained from crossing the saturated mixing ratio lines with the vertical profile of the air temperature. This moisture increase is especially true for the central-southern region (site B; Fig. 8c) where there is more precipitable water (PW values at the top of Fig. 8a and c). In summer,...

to

"...Even though the Skew-T diagram indicates that the relative humidity is rather low, the warmer atmosphere results in a small increase of moisture availability in the middle-to-low atmosphere (water vapour) in winter. Note that the higher moisture availability is illustrated by the increase of the values of the mixing ratio, which are obtained from crossing the saturated mixing ratio lines with the vertical profile of the dew-point temperature. This slight moisture increase is found in the central-southern region (site B, Fig. 8c) where there is more precipitable water (PW values at the top of Fig. 8c). In summer,..."

**13.** *Page 12, lines 364-365: Again, the actual mixing ratio is given by dew point line crossing with saturated mixing ratio*

**RESPONSE:**

We changed it in the revised manuscript according to the referee's suggestion.

We would like to thank the referee for the time invested in reviewing our manuscript so carefully. We look forward to meeting the referee's expectations.

Best regards,

Patricio Velasquez (on behalf of the author team)

**References**

Bereiter, B., Fischer, H., Schwander, J., and Stocker, T. F.: Diffusive Equilibration of $N_2$, $O_2$ and $CO_2$ Mixing Ratios in a 1.5-Million-Years-Old Ice Core, The Cryosphere, 8, 245–256, https://doi.org/10.5194/tc-8-245-2014, 2014.

Berger, A.: Long-Term Variations of Daily Insolation and Quaternary Climatic Changes, Journal of the Atmospheric Sciences, 35, 2362–2367, https://doi.org/10.1175/1520-0469(1978)035<2362:LTVODI>2.0.CO;2, 1978.

Hofer, D., Raible, C. C., Dehnert, A., and Kuhlemann, J.: The Impact of Different Glacial Boundary Conditions on Atmospheric Dynamics and Precipitation in the North Atlantic Region, Climate of the Past, 8, 935–949, https://doi.org/10.5194/cp-8-935-2012, 2012a.

Hofer, D., Raible, C. C., Merz, N., Dehnert, A., and Kuhlemann, J.: Simulated Winter Circulation Types in the North Atlantic and European Region for Preindustrial and Glacial Conditions: Glacial Circulation Types, Geophysical Research Letters, 39, L15 805, https://doi.org/10.1029/2012GL052296, 2012b.

Kjellström, E., Brandefelt, J., Näslund, J.-O., Smith, B., Strandberg, G., Voelker, A. H. L., and Wohlfarth, B.: Simulated Climate Conditions in Europe during the Marine Isotope Stage 3 Stadial, Boreas, 39, 436–456, https://doi.org/10.1111/j.1502-3885.2010.00143.x, 2010.

Ludwig, P., Pinto, J. G., Raible, C. C., and Shao, Y.: Impacts of Surface Boundary Conditions on Regional Climate Model Simulations of European Climate during the Last Glacial Maximum, Geophysical Research Letters, 44, 5086–5095, https://doi.org/10.1002/2017GL073622, 2017.

Schilt, A., Baumgartner, M., Blunier, T., Schwander, J., Spahni, R., Fischer, H., and Stocker, T. F.: Glacial–Interglacial and Millennial-Scale Variations in the Atmospheric Nitrous Oxide Concentration during the Last 800,000 Years, Quaternary Science Reviews, 29, 182–192, https://doi.org/10.1016/j.quascirev.2009.03.011, 2010.

Strandberg, G., Brandefelt, J., Kjellstro M., E., and Smith, B.: High-Resolution Regional Simulation of Last Glacial Maximum Climate in Europe, Tellus A: Dynamic Meteorology and Oceanography, 63, 107–125, https://doi.org/10.1111/j.1600-0870.2010.00485.x, 2011.

Velasquez, P., Messmer, M., and Raible, C. C.: A New Bias-Correction Method for Precipitation over Complex Terrain Suitable for Different Climate States: A Case Study Using WRF (Version 3.8.1), Geoscientific Model Development, 13, 5007–5027, https://doi.org/10.5194/gmd-13-5007-2020, 2020.

Velasquez, P., Kaplan, J. O., Messmer, M., Ludwig, P., and Raible, C. C.: The Role of Land Cover in the Climate of Glacial Europe, Climate of the Past, 17, 1161–1180, https://doi.org/10.5194/cp-17-1161-2021, 2021.

---

## Author Response (AR3)

**Response to Editor's comments**

We greatly appreciate the Editor Dr. Qiuzhen Yin for the time invested in reading the manuscript.

Regarding the Editor's question, we double checked the values within the models and we agree that the value of angular precession is indeed 195.22 for MIS4. Therefore, we included this value in the Table 1 in the revised version of the manuscript. We thank the Editor very much for identifying this inconsistency.

Best regards,

Patricio Velasquez (on behalf of the author team)